
# Measuring compound flood potential from river discharge and storm surge extremes at the global scale and its implications for flood hazard

Anaïs Couasnon[1], Dirk Eilander[1,2], Sanne Muis[1,2], Ted I.E. Veldkamp[1,5], Ivan D. Haigh[3], Thomas
Wahl[4], Hessel Winsemius[2,6], Philip J. Ward[1]

[1]Institute for Environmental Studies (IVM), Vrije Universiteit Amsterdam, De Boelelaan 1087, 1081 HV Amsterdam, The Netherlands
[2]Deltares, P.O. Box 177, 2600 MH Delft, The Netherlands
[3]School of Ocean and Earth Sciences, National Oceanography Centre, University of Southampton, Southampton, United Kingdom.
[4]Department of Civil, Environmental, and Construction Engineering and National Center for Integrated Coastal Research, University of Central Florida, United States.
[5]International Institute for Applied Systems Analysis (IIASA), Laxenburg, Austria
[6]Department of Water Management, Faculty of Civil Engineering and Geosciences, Delft University of Technology, Delft, Netherlands

*Correspondence to*: Anaïs Couasnon (anais.couasnon@vu.nl)

**Abstract.** The interaction between physical drivers from oceanographic, hydrological, and meteorological processes in coastal areas can result in compound flooding. Compound flood events, like Cyclone Idai and Hurricane Harvey, have revealed the devastating consequences of the co-occurrence of coastal and river floods. A number of studies have recently investigated the likelihood of compound flooding at the continental scale based on simulated variables of flood drivers such as storm surge, precipitation, and river discharges. At the global scale, this has only been performed based on observations, thereby excluding a large extent of the global coastline. The purpose of this study is to fill this gap and identify potential hotspots of compound flooding from river discharge and storm surge extremes in river mouths globally. To do so, we use daily time-series of river discharge and storm surge from state-of-the-art global models driven with consistent meteorological forcing from reanalysis datasets. We measure the compound flood potential by analysing both variables with respect to their timing, joint statistical dependence, and joint return period. We find many hotspot regions of compound flooding that could not be identified in previous global studies based on observations alone, such as: Madagascar, Northern Morocco, Vietnam, and Taiwan. We report possible causal mechanisms for the observed spatial patterns based on existing literature. Finally, we provide preliminary insights on the implications of the bivariate dependence behaviour on the flood hazard characterisation using Madagascar as a case study. Our global and local analyses show that the dependence structure between flood drivers can be complex and can significantly impact the joint probability of discharge and storm surge



extremes. These emphasise the need to refine global flood risk assessments and emergency planning to account for these potential interactions.

## 1 Introduction

Flooding in deltas and estuaries is driven by the interactions of oceanographic, hydrological, and meteorological phenomena such as extreme rainfall, river discharge, storm surge, and wave action. When these co-occur in space and time, they can exacerbate the flood extent, depth, and duration locally, resulting in a so-called compound flood event (Zscheischler et al., 2018). These events have the potential to cause large social and economic impacts, and can directly or indirectly impact flood emergency response and infrastructure (Leonard et al., 2014; Zscheischler et al., 2018). The potentially extreme impacts of compound flood events were recently highlighted by Cyclone Idai in March 2019. The long-lived tropical cyclone travelled within the Mozambique Channel causing catastrophic damage along its path in Madagascar, Malawi, Zimbabwe, and most importantly in Mozambique where it made landfall twice. Its second landfall near Beira on March 14 devastated the city and surrounding villages. The combination of extreme winds of more than 160 km/h with torrential rains happening several days prior to and after the landfall contributed to the severe and widespread flooding observed in this area. At this moment, there are no observations of sea levels freely available; maximum storm surge of 4.4 m were calculated for this location and more than 600 mm of accumulated rainfall were measured over a period of two weeks, resulting in local rivers going out of their bank (JRC, 2019; Probst and Annunziato, 2019). Preliminary estimates report that Idai directly affected 3,000,000 people, killed at least 1,000 people, destroyed about US$1 billion in infrastructure, ruined 500,000 hectares of crops, and caused widespread power outages, and multiple road closures that complicated aid distribution and ongoing humanitarian interventions to keep cholera outbreaks under control (Bloomberg, 2019; CBC, 2019; ERCC, 2019). In Europe, between 1870 and 2016 at least 23 damaging flood events reported the co-occurrence of coastal and river floods, representing about 1.5% of all flood events (Paprotny et al., 2018b). For this study, Paprotny at al. (2018b) used four indicators (i.e.: losses, persons affected, persons killed or area flooded) as a threshold to consider an event damaging. However, little is known about the impacts and occurrence of compound flood events globally (Bevacqua et al., 2017).

Classical methodologies for large-scale flood hazard and flood risk studies mainly consider univariate flooding mechanisms and do not include compound flood events (Ward et al., 2015). These assessments therefore focus on either river (e.g. Alfieri et al., 2014; Dottori et al., 2016; Hirabayashi et al., 2013; Ward et al., 2013, 2017, Winsemius et al., 2013, 2016) or coastal floods (e.g. Brown et al., 2016; Hinkel et al., 2014; Muis et al., 2016; Vousdoukas et al., 2018), neglecting riverine and sea level interactions. Yet, these interactions can significantly influence the magnitude of simulated water levels. At the global scale, Ikeuchi et al. (2017) found the annual maximum river water level for 2005 in low-lying flat areas to be underestimated by more than 0.5 m when ignoring sea level interactions. Local studies have shown that ignoring the dependence between river discharge and storm surge can underestimate the return period of a given water level within a river mouth (Bevacqua et al., 2017; Couasnon et al., 2018; Moftakhari et al., 2019; Serafin et al., 2019).



Compound flood events can occur due to synoptic weather systems (Seneviratne et al., 2012). Clearly, tropical cyclones have the potential to cause simultaneous high river discharge and storm surge, as exemplified by Cyclone Idai. Storms with prevailing wind directions hitting mountains have also been documented to generate strong sustained winds accompanied with intense rainfall due to orographic effects (Martius et al., 2016; Svensson and Jones, 2002, 2004). However, the co-

occurrence of coastal and river floods can also occur by chance and not be related to an underlying common synoptic weather system. The expected number of co-occurrences happening by chance (i.e., under statistical independence) can be determined based on probability theory (Kew et al., 2013; Martius et al., 2016). The impact of a compound flood event is influenced by the magnitude of the river and coastal flood drivers. The presence of a positive and significant statistical dependence between flood drivers indicates a higher probability for the occurrence of extreme combinations of these

variables when compared to statistical independence (Diermanse and Geerse, 2012).

A consistent mathematical definition of compound flood events does not exist and multiple statistical methods have been suggested to study this phenomenon (Hao et al., 2018). These methods usually examine the number of joint extremes or the statistical dependence between proxy variables of different flood hazard types such as rainfall and storm surge, river flow and storm surge, and river flow and sea level (Bevacqua et al., 2018; Hendry et al., 2019; Kew et al., 2013; Paprotny et al.,

2018a; Svensson and Jones, 2002, 2004; Wahl et al., 2015; Ward et al., 2018; Wu et al., 2018; Zheng et al., 2013). Recent compound flooding studies carried out at the regional to global scale used copula theory to characterise the bivariate joint distribution and assess complex dependence structures, for example in the case of upper tail dependence (Bevacqua et al., 2018; Paprotny et al., 2018a; Ward et al., 2018). Possible compound flooding mechanisms are examined by sampling a set of events from the full bivariate time-series, and then analysing the dependence structure of the latter. Wahl et al. (2015),

Moftakhari et al. (2017) and Ward et al. (2018) used conditional sampling to assess the bivariate relationship between a riverine flood driver and a coastal flood driver when one variable was in an extreme state (for example by selecting annual maxima or peaks over threshold). Other studies defined compound flood events as pairs based on joint exceedances above a predefined quantile such as the 95[th] or 97.5[th] percentile of the respective marginal distribution (Bevacqua et al., 2018; Hawkes, 2008; Kew et al., 2013). However, directly applying such approaches for flood hazard quantification can be

difficult due to the challenge of both defining independent and identically distributed events, and capturing extremes from both time-series (Hawkes, 2008; Hawkes et al., 2008).

A statistically robust analysis of bivariate flood drivers requires an extensive set of high-quality observations. Studies based on observations from gauge data have therefore provided an overview of the compound flood potential globally, but are strongly biased towards gauge-rich areas. In the case of Ward et al. (2018), this resulted in a selection of 187 pairs of stations

located mainly around the coasts of North America, Europe, Australia, and Japan. Non-stationarities in the observations may be present due to anthropogenic activities, such as water extractions, dam construction, and land-use changes. These factors increase the complexity of the signal and make the attribution of the dependence to synoptic meteorological drivers challenging. One way to address these limitations is by using hydrodynamic models to simulate river discharge and storm surge, and using these simulated time-series for the statistical analysis of compound flood potential. Such an approach has



been carried out for the European (Bevacqua et al., 2018; Paprotny et al., 2018a) and the Australian coastlines (Wu et al., 2018).

In this paper, we identify hotspot regions for compound floods from riverine and coastal floods along the entire global coastline by taking advantage of the extensive spatial and temporal coverage from a global river discharge and a global

storm surge model. In doing so, we provide a first statistical assessment of the compound flood potential in areas where observations from discharge and tide gauges are absent or insufficient. We do not limit our analysis to one specific statistical approach, but purposefully examine the compound flood potential by analysing both the timing between river discharge and storm surge extremes, and their dependence. We further suggest and apply a new methodology to quantify compound flood hazard that integrates these characteristics while fully capturing both extreme marginal distributions. Finally, we exemplify

the critical influence of the dependence structure on the probability of compound discharge and coastal flood events by means of a case study example in Madagascar. Therefore, our global analysis should be considered as a first step towards statistically characterising compound flooding from extreme river discharge and storm surge worldwide.

This paper is divided in four parts, as follows. Section 2 introduces the global datasets used and the method developed for this study. Section 3 presents the results and discusses the observed spatial patterns of high (low) compound flood potential

based on previous literature. We also emphasise the implication of compound flood events for flood hazard quantification by looking at a selected location in Madagascar. The limitations of our study, as well as the conclusions and outlook for future research are presented in Section 4.

## 2 Data and Methods

We assess the compound flood potential between riverine and coastal flood drivers using simulated daily river discharge and

maximum daily storm surge as proxy variables, respectively. The latter is a common choice for studying compound flood hazard analysis in deltas and estuaries (Khanal et al., 2018; Klerk et al., 2015; Svensson and Jones, 2002; Ward et al., 2018). The research involves the following steps, each of which is described in the following subsections:

1.  Selecting global datasets of river discharge and storm surge time-series;
2.  Defining sets of events to analyse compound flooding; and
3.  Quantifying compound flood potential using the defined sets.

### 2.1 Selecting Global Datasets of River Discharge and Storm Surge Variables

We use simulations of instantaneous daily discharge of the CaMa-Flood model v362 (Yamazaki et al., 2014) forced by daily average runoff data of the JULES model WRR2 eartH2Observe (Best et al., 2011; Clark et al., 2011; Dutra et al., 2017; Schellekens et al., 2017) available at: https://doi.org/10.5281/zenodo.3258007. The maximum daily storm surge is obtained

from the Global Tide and Surge Model (GTSM) (Muis et al., 2016; Verlaan et al., 2015). These two datasets are selected because they have shown good performance when compared to outputs from other global-scale models and are in good



agreement with observations (Beck et al., 2017b; Muis et al., 2016; Schellekens et al., 2017). Both models were forced based on the same meteorological dataset, namely the ERA-Interim global reanalysis dataset developed by the European Centre For Medium-Range Weather Forecasts (Dee et al., 2011). For precipitation, the MSWEPv1.2 dataset was used, which complements the ERA-Interim dataset with other reanalysis, satellite, and gauge datasets (Beck et al., 2017b). In the

following paragraphs, we provide an overview of both global models.

Daily river discharge is obtained by routing the mean daily runoff of the JULES model from the eartH2Observe WRR2 reanalysis data at 0.5° resolution (Best et al., 2011; Clark et al., 2011; Schellekens et al., 2017) with CaMa-Flood at a 0.25° resolution (Yamazaki et al., 2011). The output is the instantaneous discharge at GMT 00:00 daily for the period 1980-2014. For the eartH2Observe WRR2 reanalysis dataset, the hydrological model was forced with temperature and potential

evaporation derived from ERA-Interim and precipitation from the MSWEPv1.2 dataset (Beck et al., 2017c). The effect of human water use on the water balance was not included, and therefore the dataset characterises the compound flood potential stemming from the climate forcing only. Additional pre-processing of the runoff data was required to define runoff and remove negative runoff outliers (Eilander et al., 2019). The river discharge obtained at the coast is based on the assumption of a constant 0 m +EGM96 coastal water level and not corrected for coastal discharges, for example due to the influence of

tidal currents, which means that the discharge variable is the result from upstream catchment processes only. The JULES model was specifically selected as it showed one of the best mean overall performances in terms of runoff signatures and temporal correlation when excluding polar regions (Beck et al., 2017b).

Storm surge, the change in sea level driven by high winds and low atmospheric pressure, is simulated in GTSM with wind speed and atmospheric pressure from ERA-Interim (Muis et al., 2016; Verlaan et al., 2015). The model is a global

hydrodynamic model using an unstructured grid with a higher resolution on the shallow continental shelf (up to 3 arc min) than in deeper parts of the oceans (0.5°). The surge component is modelled separately from the tide and thereby does not include surge-tide interactions, which allows us to isolate the meteorological contribution only. Storm surge time-series are available at 16,395 output locations unevenly distributed along the coastline, with a temporal resolution of 10 minutes between 1979-2014.

We carry out additional validation for both models extending the validation performed by Beck et al. (2017b) and Muis et al. (2016), by looking more specifically at the timing and correlation of discharge and storm surge extremes over a time period of at least 20 years between 1980-2014, see Supplement S1. We calculate the percentage of annual maxima dates correctly predicted and the Spearman's rank correlation coefficient between observed and simulated annual maxima. For the discharge, we find a relatively high rank correlation globally (median: 0.56, s.d.: 0.23). Capturing the timing of extreme river

discharge is more challenging (median hit rate: 0.21, s.d.: 0.18), but the hit rate increases close to the coast (see Figure S1 in Supplement S1). For the storm surge, we find a higher hit rate (median: 0.32, s.d.: 0.21) and a lower rank correlation coefficient (median: 0.37, s.d.: 0.31) than for the discharge. In this case, coastal stations with a high correlation coefficient also capture well the timing of storm surge extremes. As a result, the timing and correlation of extreme storm surge is generally well represented along the European, North American, Japanese, and Australian coast and less well captured for



the South African and South American coasts. Therefore, even though the performance of both models varies globally, it provides an acceptable performance on average for the purpose of this study. i.e. to provide a first-cut assessment of the compound flood hazard potential at the global scale.

Finally, each discharge location at the river mouth of coastal catchments larger than 1,000 km² is paired with the nearest 5  ($\leq$ 75 km) GTSM output location (Eilander et al., 2019). This leads to 3,979 pairs of river discharge and storm surge time-series between 1980 to 2014, representing 35 years of daily data.

## 2.2 Defining Sets of Events to Analyse Compound Flooding

We do not restrict our analysis to one specific set of extreme river discharge and storm surge events per location, but instead define different sets of events from the paired time-series in order to measure the compound flood potential (presented in 10  Section 2.3). In this subsection, we explain the differences between the sets and illustrate them for an example location along the Mexican coast (Fig. 1c). Figure 1a and 1b present the paired time-series of simulated daily discharge $q$ and the maximum daily storm surge $s$ for the example location.

To investigate the strength of the dependence between the two variables, we select the conditional sampling method used in Wahl et al. (2015) and Ward et al. (2018). We create two sets of events based on the conditional sampling of the annual 15  maxima of the river discharge $Q$ and storm surge $S$. We select for year $n$ the maximum of the daily storm surge height $s_n$ within $t_n \pm \Delta$ days from the occurrence of the annual maximum of the river discharge $Q_n$:

$$s_n = \max(s^{t_n-\Delta}, \dots, s^{t_n+\Delta}) \qquad \text{where } t(Q_n) = t_n \qquad (1)$$

Conversely, the other set is created as follows:

$$q_n = \max(q^{t_n-\Delta}, \dots, q^{t_n+\Delta}) \qquad \text{where } t(S_n) = t_n \qquad (2)$$

This leads to two sets of pairs $(Q_n, s_n)$ and $(S_n, q_n)$ with $n = 1, 2, \dots, 35$. The two sets of events are shown in Figure 1d for 20  the example location and a time window of $\Delta = 3$ days. They can be interpreted as the highest daily storm surge height (river discharge) associated with the river discharge (storm surge) annual maximum. Note that peaks could also be selected based on a peaks over threshold (POT) approach. We do not expect this choice to significantly influence the results if selecting an equivalent threshold, as investigated by Ward et al. (2018).

We also examine the co-occurrence of annual maxima by defining another set of events, the annual maxima pairs of river 25  discharge and of storm surge: $(Q_1, S_1), \dots, (Q_n, S_n)$. If the timing between both annual maxima in a year is less than or equal to $\Delta$ days, i.e.: $|t_{Q_n} - t_{S_n}| \leq \Delta$, we consider it as a co-occurring event. We denote such a co-occurrence by $(Q_n^*, S_n^*)$. Figure 1e shows all the pairs of annual maxima obtained for the example location for $\Delta = 3$ days. In this case, three co-occurring events are recorded over the whole 35 years of simulation period (red dots). We transform the annual maxima pairs to probability space using their respective empirical cumulative distribution functions (Figure 1f). The co-occurring events




(shown in red) do not correspond to joint high quantiles but a combination of high, moderate, and low storm surge with moderate to high quantiles of discharge. Unlike the conditional sampling method, the marginal distribution using this sampling approach now corresponds to the respective annual maxima distribution. This means that we can easily convert the corresponding quantiles to their marginal return period.

5   In the extreme case where annual maxima of discharge and surge are always co-occurring, this means that all the sets of events defined above are equivalent. In other words, $(Q_n, s_n) = (S_n, q_n) = (Q_n, S_n) = (Q_n^*, S_n^*)$. We do not expect to observe such an extreme case, but this highlights that $(Q_n^*, S_n^*)$ events are always part of all sets. For the example location, the three overlapping pairs from both conditional sets in Figure 1d correspond to the co-occurring annual maxima in Figure 1e and 1f.

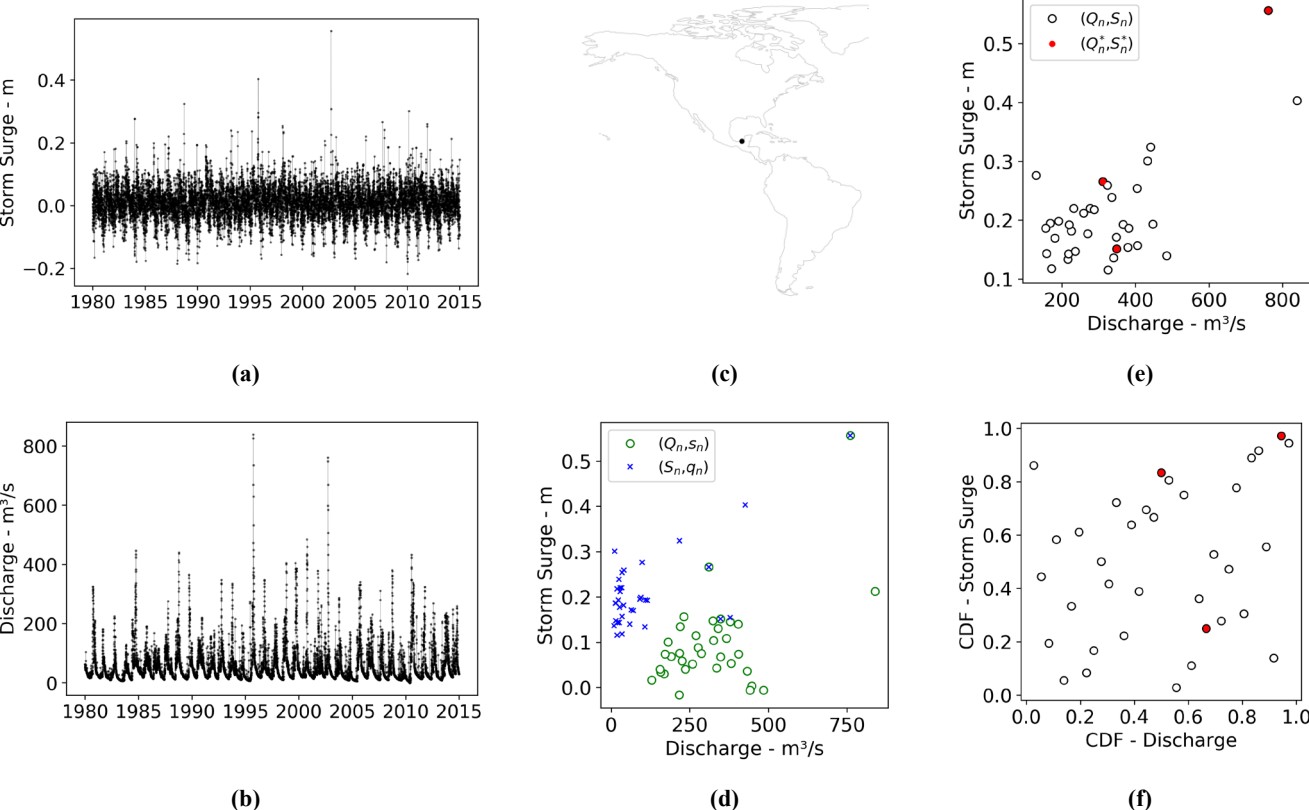

**Figure 1: Maximum daily storm surge, *s*, (a) and daily discharge , *q*, (b) paired for a location along the Mexican coast (c). Discharge *Q* and storm surge *S* annual maxima from both (a) and (b) are used to construct event time-series based on the conditional sampling method (d) and joint annual maxima (e), using a time window of Δ = 3 days. Events shown in (e) are shown in probability space in (f).**


### 2.3 Quantifying Compound Flood Potential using the Defined Sets

We use the sets of events defined in Section 2.2 to measure the compound flood potential at all the paired locations in three ways. First, we calculate the conditional dependence strength between river discharge and storm surge. Second, we calculate the total number of co-occurring annual maxima from the simulation period and analyse the probability of obtaining such a result if discharge and surge were independent. Finally, we calculate the compound flood hazard that corresponds to the probability of observing a co-occurring discharge and storm surge annual maxima above a certain magnitude in a given year. These three approaches are described in the three sub-sections below.

### 2.3.1 Conditional Dependence Strength

We characterise the interactions between river discharge and storm surge by calculating the rank correlation coefficient for the two sets of pairs $(Q_n, s_n)$ and $(S_n, q_n)$ constructed from the conditional sampling method at all paired locations. We use Spearman's rank correlation coefficient $r_s$ to assess the monotonic dependence strength. This is an advantage over the Pearson's linear correlation coefficient, which quantifies the presence of linear relationships. We present results for values with a statistical significance up to 5% level ($\alpha = 0.05$) and a time period of $\Delta = 3$ days, but perform a sensitivity analysis of $\Delta$ for up to 7 days and $\alpha = 0.10$ (Supplement S3). For the example in Figure 1d, the dependence patterns observed correspond to $r_s = 0.22$ (p-value: 0.20) for the $(Q_n, s_n)$ pairs and $r_s = 0.28$ (p-value: 0.10) for the $(S_n, q_n)$ pairs. This different dependence behaviour suggests a higher compound flood potential for discharge conditional on storm surge extremes than in the other case, but are not statistically significant. This is further analysed in Section 3.1.

### 2.3.2 Number of Co-occurring Annual Maxima

We analyse the number of simulated co-occurrences of annual maxima of river discharge and storm surge. To do so, we calculate the total number $X$ of co-occurring annual maxima $(Q_n^*, S_n^*)$ obtained for each paired location over the whole simulation period. Let $x$ represent the total number of co-occurrences within $N$ years (here $N = 35$ and $x = 0,1,...,35$), we use a binomial distribution to calculate the probability of obtaining $x = X$ co-occurrences under independence:

$$P(x = X) = \binom{N}{X} p^X (1-p)^{N-X} \tag{3}$$

where $p$ is the probability of a co-occurrence in a given period under independence. We empirically derive $p$ by assuming that the co-occurrence can happen randomly within a period of a year (365 days) or three months (90 days), based on 1,000,000 repetitions and for different values of the time window $\Delta$. For example, for $\Delta = 3$ days, we find $p = 0.0187$ for the former and $p = 0.0760$ for the latter.


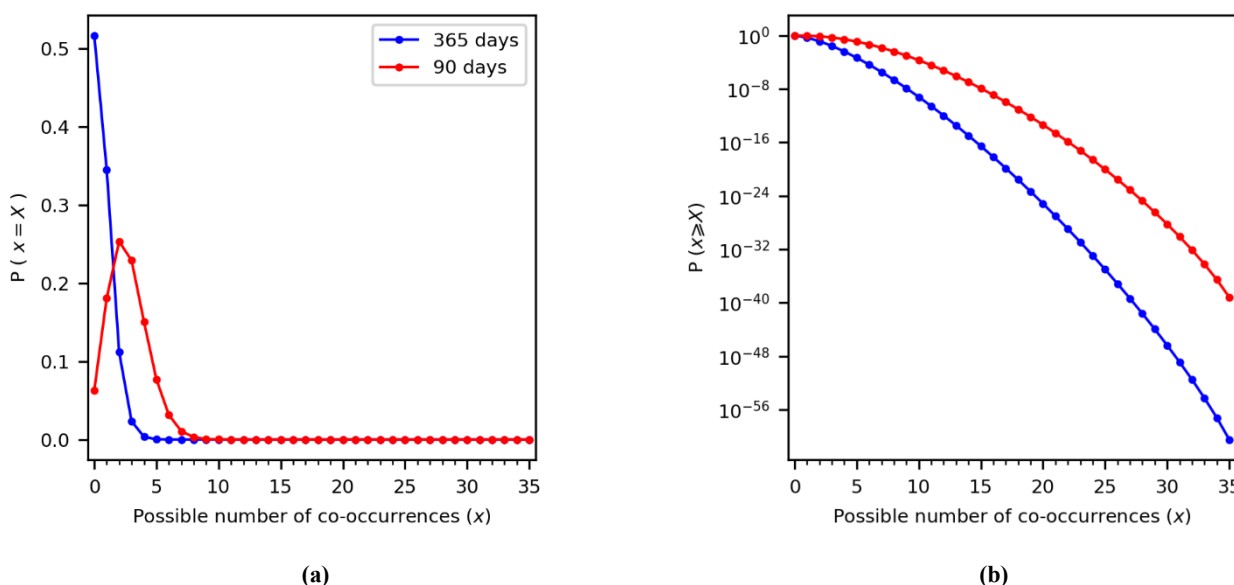

(a)        (b)

**Figure 2: Probability mass function for the probability of observing $X$ co-occurrences of annual maxima in N=35 years and for $\Delta = 3$ days (a). Same as (a) but shown as the exceedance probability $P(x \geq X)$ (b).**

As shown in Figure 2a, one can expect a higher chance of observing 2 or more co-occurrences within the 35 years if both annual maxima are consistently occurring within the same 90 days season (red curve) compared to if they occur randomly throughout the whole year (blue curve). This finding can be summarised as the exceedance probability of obtaining $X$ or more co-occurrences $P(x \geq X)$, and is calculated as the area under the curve right of $x = X$ (Figure 2b). As the number of co-occurrences increases, the probability of observing such a situation in the data due to randomness decreases for all cases and approaches 0 (Figure 2b), but this value is modulated by the period in which both annual maxima can occur. For our example location, we observe $x = 3$ co-occurrences. Assuming that both annual maxima occur randomly within the same season of 90 days, we have $P(x \geq X) = 0.50$, whereas if they can occur randomly within the year, $P(x \geq X) = 0.03$. These distributions also show that one would expect to observe some locations with a large number of co-occurrences even under independence (right tail of the distributions), but these situations are expected up to a certain frequency. Therefore, we compare the distributions of co-occurrences obtained with the ones shown in Figure 2.

### 2.3.3 Quantification of the Compound Flood Hazard

Finally, we examine the probability $P_F$ of observing a co-occurring riverine and coastal event in any given year with a discharge magnitude and a storm surge magnitude higher than a threshold value $z_1, z_2$, respectively. We refer to probability $P_F$ as the compound flood hazard and calculate it as follows:

$$P_F = P(Q > z_1 \cap S > z_2) \times P_c \qquad (4)$$

where $P_c$ is the probability of a co-occurrence in a given year. We estimate $P_c$ from the number of co-occurring annual maxima obtained within the simulated time-series. Here, we assume that $P_c$ is not a function of the threshold considered and


is equally distributed over the probability space. This assumption seems reasonable based on visual observations at randomly selected locations (see Supplement S2). Therefore, at a given paired location:

$$P_c = \frac{X}{N} \tag{5}$$

If no co-occurrences were measured ($X = 0$), we select $P_c = p$, i.e. the probability of observing a co-occurrence under independence in any given year. As explained in Section 2.3.2, this requires some knowledge about the coastal and riverine flood season. Here, we simplify the analysis by selecting a flood season of 365 days. The joint survival probability, $P(Q > z_1 \cap S > z_2)$, can be quantified as follows using copula modelling (Serinaldi, 2015):

$$P(Q > z_1 \cap S > z_2) = 1 - u - v + C(u, v) \tag{6}$$

where $C$ is the copula function joining the uniform ranks $u, v$ of variables $Q$ and $S$, respectively. We might underestimate the joint probability $C(u, v)$ if the strength of the dependence between the $(Q, S)$ pairs significantly deviates from the $(Q^*, S^*)$ pairs. We use bootstrapping to calculate whether the correlation between co-occurring annual maxima is statistically different to the correlation between non co-occurring annual maxima (two-tailed test, significance level α = 0.05). Finally, if no statistical dependence is measured, we assume independence between the magnitude of the joint exceedances and equation 6 reduces to the following product:

$$P(Q > z_1 \cap S > z_2) = (1 - u) \times (1 - v) \tag{7}$$

Given the limited temporal coverage of the data, we present the result for a relatively low threshold value, a quantile threshold equivalent to a 5-year discharge magnitude and a 5-year storm surge magnitude (i.e., $u = v = 0.8$). We also select the Gaussian copula to model the dependence structure, but assess the sensitivity of this choice on the compound flood hazard, using Madagascar as a case study.

## 3 Results and Discussion

In this section, we present the results for each compound flood potential measure introduced in Section 2.3 along the global coastline. We compare the results with respect to existing literature on compound flooding. Relevant meteorological processes likely to lead to the observed regional patterns of high compound flood potential are also discussed.

### 3.1 Conditional Dependence Strength

Figure 3 presents the Spearman's correlation coefficient $r_s$ for all paired locations along the global coastline and a time window of Δ = 3 days around the flood annual maxima. For storm surge conditional on extreme discharge, $(Q_n, s_n)$ pairs, we find statistically significant (α = 0.05) and positive dependence for 14% of locations (Fig. 3a). For discharge conditional on extreme storm surge, $(S_n, q_n)$ pairs, we find statistically significant and positive dependence for 19% of locations (Fig. 3b). On average, the dependence is also slightly stronger for the latter case compared to the former (mean $r_s = 0.10$ for $(Q_n, s_n)$ pairs; mean $r_s = 0.11$ for $(S_n, q_n)$ pairs; Welch's t-test, p-value: 0.007). This suggests some higher chance to have





a high discharge when there is also an extreme storm surge than vice versa. Finally, 74% of the locations do not exhibit statistically significant correlation for either case. A similar analysis was performed by varying the time window $\Delta$ from 0 until 7 days (Table S1 in Supplement S3) and was found to lead to similar results, except for $\Delta = 0$ days where we observe a smaller value of 11% for $(Q_n, s_n)$ pairs. For a higher significance level ($\alpha = 0.10$), a higher percentage of statistically significant correlations is found (17%-22% for $(Q_n, s_n)$ pairs; 24% for $(S_n, q_n)$ pairs), but the results are consistent with those obtained under $\alpha = 0.05$.

We observe clear regional patterns of positive dependence globally. These dependence behavior patterns are similar to those found by Ward et al. (2018) using observations from river and tide gauges. We obtain more locations for South-western Japan exhibiting statistically significant dependence when conditional on extreme storm surge (Fig. 3b) than when conditional on extreme discharge (Fig. 3a). We also find positive and statistically significant dependence for locations both on the West and East Coast of the United States (US). However, our results also highlight regions that could not be examined by Ward et al. (2018) due to a lack of gauge observations. Along the South American coastline, we find a cluster of positive dependence along the South coast of Chile. Along the African coastline, the coast of Madagascar is consistently highlighted in both cases, as well as the coast of Morocco. Finally, the coasts of India and large parts of East Asia also show large regions with positive dependence.

Other regions, such as the East coast of the US, Italy, the United Kingdom (UK) or China, show a more complex dependence behavior. Riverine flooding in these regions is related to multiple mechanisms, not all of which are related to the mechanisms causing high storm surge. On the East coast of the US, even though tropical cyclone activity is known to contribute to high storm surge levels and intense precipitation (Villarini and Smith, 2010; Wahl et al., 2015), other river flood generating mechanisms also play a role. For example, high river discharge could also be due to snowmelt or convective storms happening upstream in the catchment (Berghuijs et al., 2016). Similarly, for the east coast of the UK, Hendry et al. (2019) found that storms that generate high river discharge are different than the ones that generate high storm surge. This explains why the presence of a statistical significance for discharge conditional on extreme storm surge (Fig. 3a) may be absent when conditionally sampling on extreme discharge (Fig. 3b).

Locations in several regional clusters exhibit a positive statistical dependence in both cases (6% of all the locations studied) and therefore present the highest potential for compound flooding. Among others, we note the coasts of Madagascar, Portugal, Northern Morocco, Northern Australia, Vietnam, and Taiwan, which all consistently show a positive and significant dependence between discharge and storm surge.
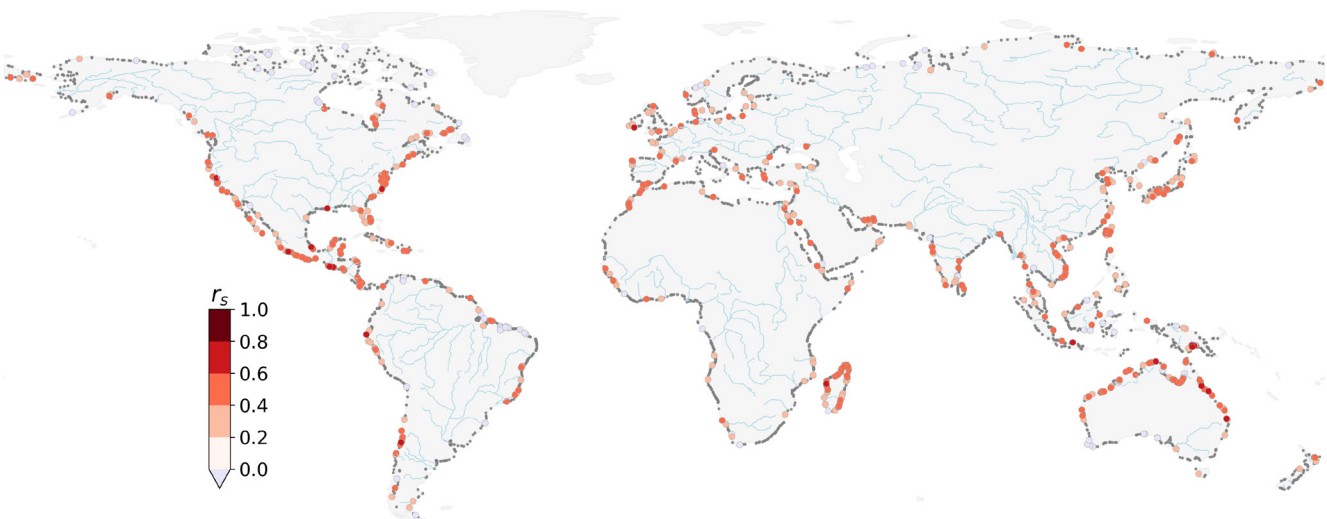

**(a)  For the set of $(Q_n, s_n)$ pairs**

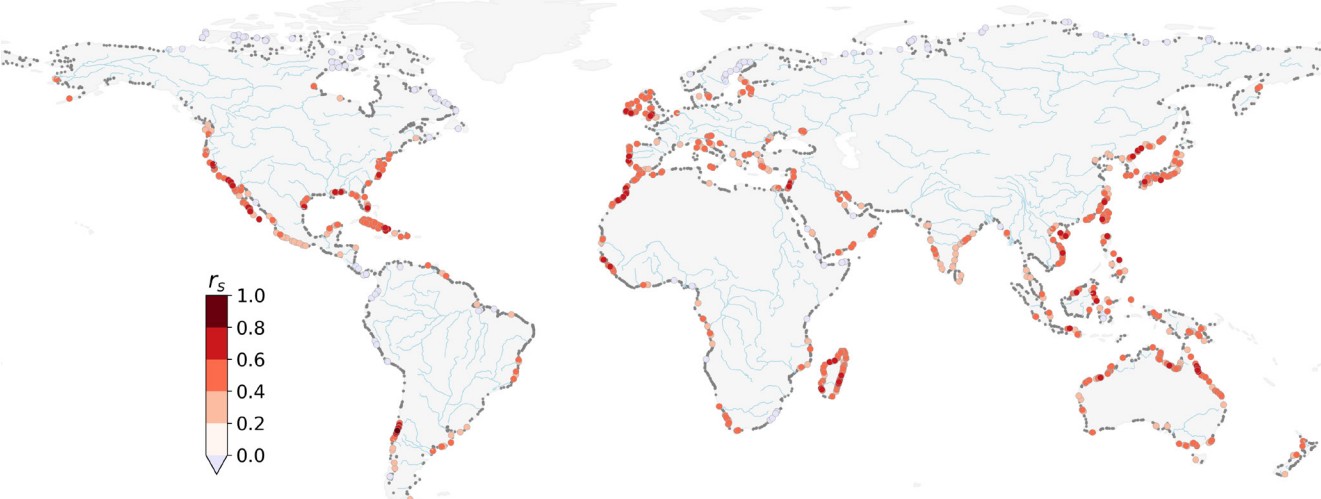

**(b)  For the set of $(S_n, q_n)$ pairs**

**Figure 3: Spearman's $r_s$ correlation coefficient between storm surge conditional on annual maxima discharge $(Q_n, s_n)$ (a), and discharge conditional on annual maxima storm surge $(S_n, q_n)$ (b). Black dots denote locations with no significant dependence ($\alpha = 0.05$). Major rivers are shown in light blue.**





## 3.2 Number of Co-occurring Annual Maxima

We examine the total number of co-occurrences between discharge and storm surge annual maxima obtained from the simulation period and assess their probability of co-occurrence under independence. Figure 4 presents the total number of simulated co-occurring annual maxima, $(Q_n^*, S_n^*)$ events, considering a time window of $\Delta = 3$ days. We observe a minimum

of 0 co-occurrence and a maximum of 19 co-occurrences within the 35 years simulation period. At 64% of the locations, we observe at least one co-occurrence. Clearly, the absence of a significant dependence (measured at 74% of the locations, see Section 3.1) does not preclude the co-occurrence of discharge and storm surge annual maxima. Finally, at 5% of the locations, yearly maxima are co-occurring more than 30% of the time (i.e. representing 10 co-occurrences or more over 1980-2014).

We test the significance of these results globally by comparing the empirical distribution obtained from the simulated data with the binomial distributions shown in Figure 2. Figure 5 presents the uncertainty bounds around the empirical distribution using non-parametric bootstrapping with 5,000 bootstrap samples and a significance level $\alpha = 0.05$. The right tail obtained in our simulated results significantly deviates from any of the binomial distributions considered here. Therefore, we obtain more simulated co-occurrences than we would expect under the assumption of independence (i.e. compared with binomial

draws). This suggests that in regions with a high number of co-occurrences, discharge and storm surge annual maxima are very unlikely to be independent and cannot be explained by seasonality only. Instead, we argue that synoptic weather systems could explain the high number of co-occurring annual maxima. We note that even though the influence of the time window clearly influences the number of co-occurrences measured (see Figure S10), it will not affect the interpretation of Figure 5. This is because the time window is already accounted for when deriving the binomial probability distributions.

Therefore, selecting a larger time window would only result in a shift of all distributions to the right, but similar conclusions would prevail.

Locations where results deviate the most from independence coincide with areas of strong tropical or extratropical cyclone activity. Martius et al. (2016) found that coastal regions affected by frequent tropical cyclones experience the highest number of co-occurring wind and precipitation extremes. Their study highlighted very similar regional patterns compared to the ones

presented in Figure 6, however with less geographical spread. This can be attributed to the fact that they focused on the analysis of climate extremes, whereas we use river discharge and storm surge where these meteorological phenomena are propagated through model chains. In other regions, they identify the interaction of weather systems with regional orographic features to cause compound wind and precipitation extremes. Atmospheric rivers landing on the Western coast of the US have caused recurrent major flood events (Gimeno et al., 2014). Composite analyses of these systems show that they can be

accompanied by extreme skew surge (Ridder et al., 2018; Ward et al., 2018). The Iberian Peninsula and the Atlas Mountains contain major orographic features that can block prevailing wind flows and trigger orographic rainfall during low pressure systems, thereby causing high river discharge. These synoptic weather systems were documented to have caused serious flood events in Portugal during windstorms Klaus in 2009, Xynthia in 2010, and Gong in 2013 (Liberato, 2014)





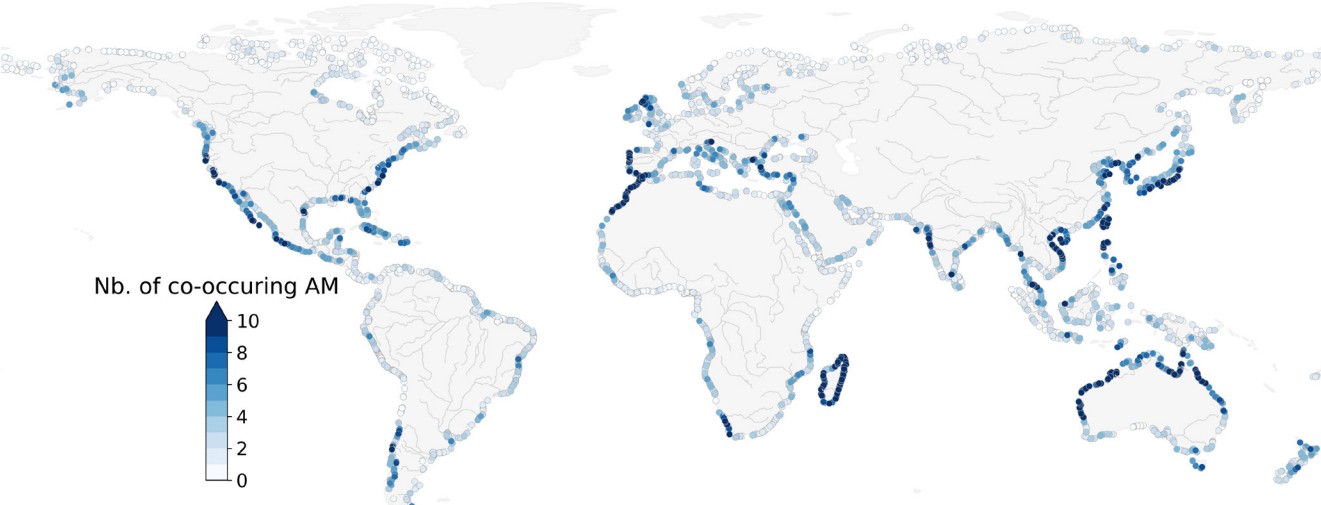

**Figure 4: Number of simulated co-occurring annual maxima of discharge and storm surge obtained between 1980-2014 using a time window of 3 days. Major rivers are shown in grey.**

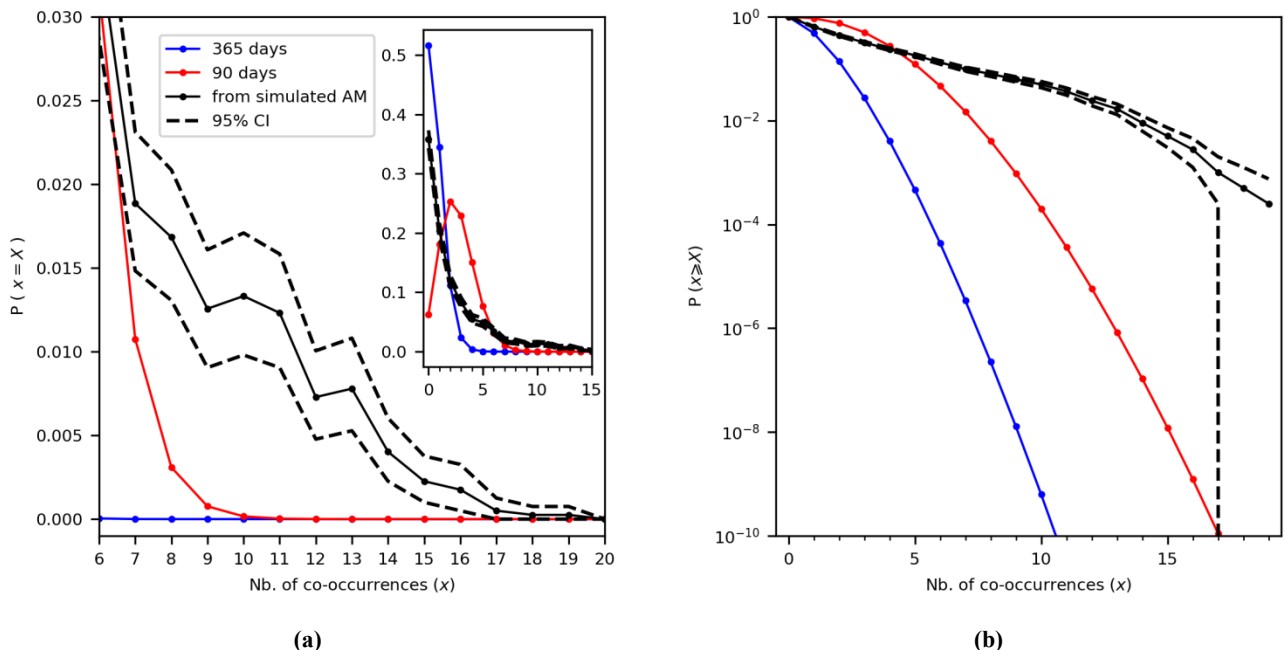

**Figure 5: Comparison of the empirical distribution of the number of co-occurrences from the simulations (in black) with the binomial distributions assuming that annual maxima can happen randomly within the year (365 days, blue curve) or within the same season (90 days, red curve). Probability mass function (a) and exceedance probability function (b). The 95% confidence intervals (CI) are calculated using bootstrapping.**



### 3.3 Quantification of the Compound Flood Hazard Potential

We use Equation 4 to calculate the probability $P_F$ of a discharge and storm surge annual maxima co-occurring in any given year with a magnitude higher than a threshold value, and referred to as the compound flood hazard potential. Figure 6 shows the result using a threshold equivalent to a 5-year return discharge level and a 5-year return storm surge level. The

probability $P_F$ is presented as a joint return period in years ($1/P_F$). A low (high) joint return period indicates a high (low) probability of a river discharge and storm surge annual maxima co-occurring, each higher than their individual 5-year return level. Assuming independence between the two variables, these conditions would be exceeded about once every 1,337 years on average (i.e.: $1/[(0.2 \times 0.2) \times 0.0187]$). In the case of complete dependence, this would happen once every 5 years on average (i.e.: $1/[0.2 \times 1]$). In 66% of the paired locations, the joint return period obtained is lower than that of

independence. Therefore, this indicates some compound flood hazard potential along most of the global coastline. However, the magnitude of this potential varies per region.

Focusing on Europe (inset in Figure 6), we find the highest compound flood hazard potential mainly on the Western coast, more specifically along the coast of Portugal, Ireland, the Western coast of the United Kingdom and the Straits of Gibraltar. This regional pattern is also observed in the studies of Bevacqua et al. (2018) and Paprotny et al. (2018a). However, contrary

to Bevacqua et al. (2018), we do not find a high compound flood hazard potential for the coast of France. We attribute this difference to the fact that we focus on river discharge whereas their study examined rainfall. In this area, a cross-correlation analysis on the results from this study (not shown here) shows a lag between ± 12 to 30 days, which exceeds the maximum lag of ± 3 days considered for this analysis.

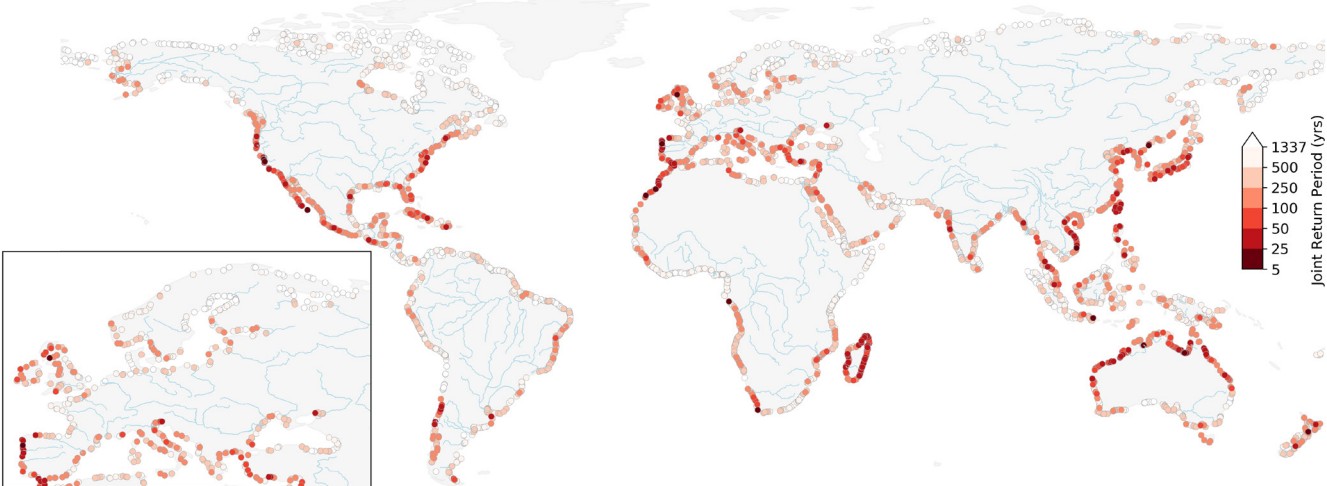

**Figure 6: Probability of a co-occurring annual maxima in a given year $P_F$ presented as the return period in years exceeding the marginal 5-year return periods of discharge and of storm surge. Major rivers are shown in light blue.**


Regions with a high compound flood hazard potential identified in Figure 6 coincide with regions with a high number of co-occurring annual maxima (Fig. 4) and a strong positive statistical dependence (Fig. 3). This is to be expected since co-occurring annual maxima $(Q_n^*, S_n^*)$ events, are present in both sets of events $(Q_n, s_n)$ and $(S_n, q_n)$. Similarly, regions with a large joint return period (500 years or more) correspond with areas with a low probability of annual maxima co-occurrence

and/or no statistical dependence. Finally, we observe regions with no clear spatial patterns, such as along the Mediterranean Sea, the Gulf of Mexico, and India. This could be explained by one or a combination of the following reasons. First, the fact that there are multiple river flood generating mechanisms that lower the likelihood of co-occurring discharge and storm surge annual maxima. Second, even in the presence of synoptic weather systems, this does not ensure a strong and positive dependence between storm surge and discharge. Drivers of maximum storm surge heights are particularly complex, and are

influenced by external factors such as local bathymetry and the geometry of the coastline (Bloemendaal et al., 2018). Third, in large catchments, there may also be a lag of several days for river flood waves to reach the basin outlet (Allen et al., 2018) such that the riverine and coastal flood annual maxima do not interact (Kew et al., 2013; Klerk et al., 2015; Ward et al., 2018).

We assess the sensitivity of the joint return period shown in Figure 6 to the selected dependence model by selecting on a

location in a region with a high compound flood hazard potential in Madagascar. The selected paired location is shown in Figure S5 in Supplement S2. Figure 7 presents the probability $P_F$ as a joint return period but for multiple dependence models, and for different threshold values corresponding to a 5-year up to a 100-year marginal return levels. We use the lowest Akaike Information Criteria (AIC) value as an indication for the best fitting bivariate copula model for the data, as implemented in the R-package *VineCopula*. Out of the 40 copula families tested, we find the Joe Clayton (BB7) copula to

best model the dependence structure. For a 5-year marginal return level, the difference in joint return period between the Gaussian and the BB7 copula models is minor (27 years and 21 years, respectively). This is not the case for higher threshold values. For threshold values corresponding to a 100-year return discharge or storm surge level, we observe an approximate fourfold increase between the two (joint return period of 1,588 years with the Gaussian copula versus 428 years with the best fitting copula model). Therefore, for large thresholds this shows that the dependence structure model can greatly influence

the probability of concurrent extremes. This is because unlike the Gaussian copula, the BB7 copula models upper tail dependence (Joe, 2015). In the presence of upper tail dependence, the dependence coefficient in the tail of the distribution is higher than the overall dependence coefficient, thereby increasing the probability of observing a concurrent extreme (Hobaek Haff et al., 2015). Even though detecting upper tail dependence with confidence from limited data length remains challenging (Serinaldi et al., 2015), these results show that it can significantly impact the joint return period. For flood

impact assessments, it is therefore recommended to thoroughly assess the dependence structure when considering multiple flood drivers.
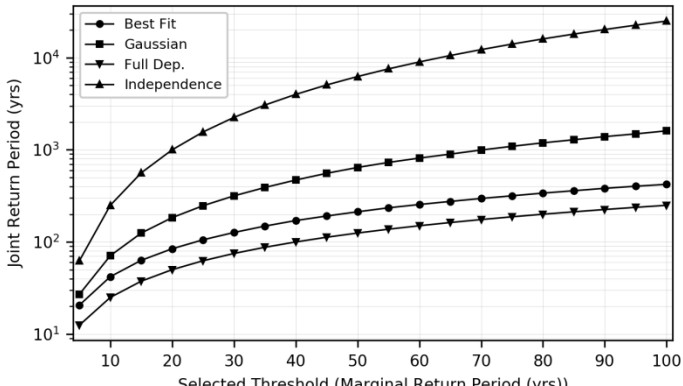

**Figure 7: Effect of the dependence structure on the joint return period of a discharge and storm surge annual maxima co-occurring in any given year with a magnitude higher than a selected threshold. The latter corresponds to the marginal return period of discharge and of storm surge levels.**

## 3.4 Limitations

While we identify compound flooding hotspot regions from extreme discharge and storm surge variables, substantial uncertainties remain as to how this phenomenon will propagate into inland flooding. Flood events in a coastal catchment typically result from the interactions of other drivers not considered in this study, such as local rainfall, wave effects, tidal amplitude, and tide-surge interactions (Arns et al., 2019; Saleh et al., 2017; Vousdoukas et al., 2016). Moreover, local characteristics like the bathymetry, catchment properties, and the presence of water control structures further influence the extent to which these interactions contribute to water level extremes at the considered river mouths (Gori et al., 2018; Veldkamp et al., 2018). Specific compound flood event modelling studies (Bilskie and Hagen, 2018; Kumbier et al., 2017) and comprehensive probabilistic simulations at local scales show that multivariate flood drivers result in highly non-linear responses of flood impact variables such as flood depth and flood extent (Couasnon et al., 2018; Serafin et al., 2019). Future assessments of compound flood hazard at the global scale should therefore focus on incorporating those multivariate processes and is left for future work.

In this study, we base our statistical analysis on annual maxima, which results in 35 data points per paired location. Alternative strategies, such as a peaks over threshold approach or sampling the *r*-largest events per year, could be used to increase the sample size (Coles et al., 2001; Tawn, 1988). However, those approaches also suffer from disadvantages (Hawkes, 2008). For example, they may result in sampling events that are not extremes and therefore underestimate the compound flood potential measured. Ward et al. (2018) found lower statistical dependence when using a POT method with a 95[th] percentile. POT methods may also result in a different number of events for the discharge than the storm surge variable if wanting to obtain independent and identically distributed extremes. This becomes particularly problematic for the analysis of their joint probability, which requires pairs. Instead, another more promising option to increase the sample size could be to work with climate ensemble models, such as those applied in Kew et al. (2013) and Khanal et al. (2018), provided that their





performance is satisfactory and that the multivariate dependence structure is not affected by bias adjustments (Zscheischler et al., 2019).

Moreover, the results presented in this study are dependent on the accuracy of the models. Even though the validation performed in this study indicated an acceptable performance on average, the accuracy of the models along large parts of the

coastline in South America, Africa, and Asia could not be assessed due to a lack of long-term gauge observations. We examined different compound flood measures and selected moderate joint return period conditions in order to allow us to compare regions and identify those most exposed to this phenomenon. Nevertheless, potentially important processes for compound flood events may be underestimated or absent in the global models used for this study. Small-scale convective and short-lived processes affecting both wind and precipitation extremes are not fully represented in the weather forcing, but

may be of critical importance in areas affected by tropical cyclones (Beck et al., 2017a; Martius et al., 2016; Muis et al., 2016). Interactions with ice and snow cover are also currently not resolved at higher latitude, which affects the timing and magnitude of both storm surge heights (Muis et al., 2016) and river discharge (Yamazaki et al., 2011). Therefore results in northern regions, where we find the lowest compound flood hazard potential, are particularly uncertain and should be interpreted with care.

Finally, we investigate compound flooding interactions under current climate conditions from hydrometeorological processes only, and neglect anthropogenic changes on the catchment and the climate. Human interventions, such as water extractions, water retention, or flood protection infrastructure can affect the travel time and magnitude of extreme discharges and modify the discharge time-series (Allen et al., 2018; Veldkamp et al., 2018). Combined with changes in environmental conditions, for example due to sea-level rise and changes in storminess, these additional non-stationary drivers can strongly

modulate the multivariate dependence structure between flood drivers and affect compound flood hazard (Moftakhari et al., 2017; Wahl et al., 2015).

## 4 Conclusions and Outlook

This paper provides a global perspective of the compound flood potential from riverine and coastal flood drivers. By selecting time series of flood drivers for both hazard types, we derived a global overview of areas particularly exposed to the

co-occurrence of high discharge and storm surge level and we quantified the strength of the interactions between the two variables. We developed compound flood potential indicators to analyse important characteristics of compound flooding related to the timing and joint dependence between river discharge and storm surge extremes. Regional clusters consistently exhibit a high potential for compound flooding. Hotspot regions such as Madagascar, Portugal, Northern Morocco, Northern Australia, Vietnam, and Taiwan all show a positive and significant dependence between flood drivers and a large number of

co-occurring annual maxima. Using the binomial distribution for different flood season lengths, we showed that the dependence between these variables cannot be explained by random extreme interactions within a season. Instead, we



hypothesise that this dependence results from synoptic weather systems and interactions between these weather systems and topography.

Extreme impact events caused by synoptic weather systems, like Hurricane Harvey or Idai, highlight the importance of considering compound flood events for flood protection in coastal communities. Currently, regulatory flood hazard maps,
such as those used in the USA, often only model flooding due to one flood driver (Federal Emergency Management Agency, 2015; Moftakhari et al., 2019). Because such a methodology discards the interactions between river and coastal floods, it can strongly flaw the representation of flood hazard in deltas and estuaries. This is also the case for current global state-of-the-art flood models, and our study provides a first indication of locations where discharge and storm surge interactions are strong. In areas coinciding with rapid economic development, this can have strong implications for emergency responders,
reinsurance, and local decision makers.

How compound flood events will affect flood impacts, adaptation strategies, and management operations at local scales is strongly dependent on local conditions, and is therefore left for future research. We presented our first insights into how the dependence structure impacts on the probability of hazardous riverine and coastal conditions globally. Such a method could be used to generate stochastic events to explore the impact of unforeseen events within a certain catchment through
hydrodynamic and impact model experiments. Similarly, future studies should investigate the importance of synoptic weather conditions with respect to the contributions from local drivers such as estuarine topography, land cover, human interventions, and water management and control in determining the impacts from current and future compound flood events.

*Data availability.* The paired daily discharge and storm surge time series at the river mouth locations used for this study are available at: https://doi.org/10.5281/zenodo.3258007. Storm surge daily maxima at all the output locations from the GTSR data set is available for scientific purposes at: https://data.4tu.nl/repository/uuid:29614991-345e-4ffd-be22-2930912a2798. High-resolution figures of the results can be found in the supplementary material and the corresponding datasets are available on request.

*Competing interests.* The authors declare no conflict of interest.

*Acknowledgments.* The research leading to these results was supported by the Netherlands Organisation for Scientific Research (NWO) in the form of a VIDI grant (grant no. 016.161.324). I.D.H was funded via the UK's National Environmental Research Council (grant no. NE/S010262/1).

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
