# Peer review of "Measuring compound flood potential from river discharge and storm surge extremes at the global scale"

_Natural Hazards and Earth System Sciences, 2019_

## Referee Comment (RC1) · Anonymous Referee #1 · 7 Aug 2019

General Comments:

Couasnon et al., present a global-scale analysis of compound flood hazard potential. Using modeled datasets of river discharge and storm surge, they highlight locations where the potential for compound flooding may have been overlooked using observational records alone, thus extending their analysis beyond that of Ward et al., 2018. They also incorporate new, creative metrics beyond conditional dependencies and copulas to describe the likelihood for co-occurrence of these variables. I feel that the research is an important contribution to the literature as it provides a global per-

spective on where compound flooding may matter for flood risk assessment and more detailed/localized studies. The manuscript is well written and describes the motivation, methods, and results very clearly, making it an overall pleasure to read. I have a few minor comments in which I believe the authors would benefit from addressing.

Specific Comments:

I appreciate that the authors consistently describe their results as "flood potential." Often the literature characterizes an extreme forcing (e.g., storm surge) as an extreme flood, without ever linking it to impacts. The authors make clear they are describing how the compounding forcing has the potential to drive flooding. On that note, I think that the title is slightly misleading/redundant, as there are really no implications for flood hazard and/or I'm not sure how that's different from "measuring compound flood potential." There are a few other locations in the text (noted in my comments below) that the authors could add "potential" to as well.

The majority of Central America, South America, and Southern Africa have a poor hit rate for storm surge (not quite as bad for discharge). Meanwhile, some of these locations have a very high number of modeled co-occurring annual maxima and correlation coefficients for (Sn,qn) (e.g., Chile and South Africa). In Section 3.4, the authors do discuss some of the limitations of being able to represent the accuracy of the model in these locations, and also warn that the results of low compound flood potential in the northern latitudes is uncertain and should be interpreted with care. Should a similar warning be provided in other locations? Do the authors have any indication they may be overestimating flood hazard potential in areas where the models do not accurately depict observations, yet have a high amount of co-occurring events?

On that note, on Page 5-6: Line 34 – Line 1, the authors state, "As a result, the timing and correlation of extreme storm surge is generally well represented . . . and less well captured for the South African and South American coasts." This statement seems to be putting it lightly, as the hit rate is 0 for all these locations, and the correlation

coefficient is negative, or relatively low. It more or less seems that it doesn't capture the South African and South American coasts.

Finally, the authors state in the conclusions (Page 18, Line 8) that, "Our study provides a first indication of locations where discharge and storm surge interactions are strong." Technically, the authors are not investigating "interactions" between these variables. Furthermore, Ward et al., 2018 provided the first indication of locations where these dependencies between variables are strong.

Comments on Specific Lines:

Page 5, Line 30: Hit rate is defined in the manuscript, but not here. Perhaps the authors could add it to the preceding sentence, "We calculate the percentage of annual maxima dates correctly predicted, termed hit rate . . ."

Page 6, Line 28: "We transform the annual maxima pairs to probability space using their respective empirical cumulative distribution functions" – I'm confused as to how this is done for 3 points (in this example at least). Am I missing something here?

Page 8, Line 19: I think simulated is sufficient when describing the datasets, and then can be removed from the rest of the text describing the datasets. The readers will know they're not observations. Anywhere else in the text makes me think there was a statistical simulation instead. "Simulated" is also used on Page 13, Line 10, caption Figure 5, and in a few other locations in the manuscript.

Page 10, Line 8: Similar to the above comment, how are the authors finding the dependence of $(Q^*, S^*)$ pairs, when there may only be 1 or 2?

Page 17, Line 20: The authors state that some approaches, "may result in sampling events that are not extremes and underestimating the compound flood potential measure." At the same time, the definition of a compound event also includes events driven by extreme and non-extreme variables, so the authors could be underestimating the types of events that could drive extreme flooding.

Page 18, Line 26: "developed compound flood potential indicators" This is the first time the authors use the word indicators, so I'm not sure which metrics they're referring to. I suggest removing this, or defining it earlier on.

Figures and Tables: Figure 1: What is (Qn, Sn) on plot 1e? This hasn't seemed to be defined. Furthermore, is there a reason the authors choose to use Mexico (Section 2.2) as the example when Madagascar is the chosen example location later?

Figure 2b: The authors may want to consider adding a zoomed inset to show what is described in the text in Line 9-11 on this page.

Figure 5: I'm a bit confused about what the black line displays. Is it the empirical distribution of all AM occurrences across the globe? If so, how is this used to describe dependence at a particular location? I assumed that the red/black line could be applied to any location, e.g., if these two variables were independent over these time scales, here's the probability it would occur. But if the black line is generated for all locations, I don't understand how it can be applied to a specific location. Am I misinterpreting this?

Supplemental Table S3: Do these numbers denote averages across all locations?

Technical Corrections: Page 6, Line 5: Better stated as, "This leads to 3,979 stations of paired river discharge and storm surge..."

Page 8, Line 2: Compound Flood potential "from extremes" (since you're not including non-extreme forcing)

Page 9, title 2.2.3 "Quantification of the Compound Flood Hazard Potential" (add potential)

Page 17, Line 6: Add "potential" e.g., ..."we identify potential compound flooding hotspots..."

If the citation Eilander et al,. 2019 has been updated to a more final format, please update citation.

---

## Referee Comment (RC2) · Anonymous Referee #2 · 13 Aug 2019

Authors explore the potential of compound flooding due to river flow and storm surge along the coast lines globally. They use numerical simulations forced by reanalysis dataset to extend our understanding about this phenomenon beyond previously reported regions with in-situ observational data. Characterization of compound flooding hazards is a very important problem in coastal regions worldwide and helps improving hazard prediction and effective resource allocation for flood risk management. The idea is interesting, study is robustly designed and manuscript is very well written. Given the fact that this is a significant contribution to the community of coastal hazard and it could

attain the readership of a broader community of natural hazard researchers, I recommend it for publication in NHESS after a minor revision. I am mainly concerned about the significance of conclusions made here, compared to previously reported patterns and results. Below, I provide more detailed comments and suggestions:

- In the abstract you mention ". We find many hotspot regions of compound flooding that could not be identified in previous global studies based on observations alone, such as: Madagascar..." and then further explore Madagascar as a case study. My question is that, given the fact that there is no observational record of discharge and storm surge in or at close proximity of Madagascar (Figures S1-S4), how reliable such compound hazard hotspot detection would be (seems among the hottest)? In other words, while with such limited information, estimation of individual extremes will be associated with significant uncertainty and errors associated with capturing the timing of extremes will add-up (Page 5), how conclusive your pattern detection would be? and why did not choose another location with more reliable record? I see you have thoroughly discussed the limitation of this work in pages 17 and 18, but still the audience needs to know the significance of results in Figure 3.

- Useful citation for Introduction Santiago-Collazo et al. (2019) A comprehensive review of compound inundation models in low-gradient coastal watersheds, Environmental Modelling & Software, Volume 119, Pages 166-181, https://doi.org/10.1016/j.envsoft.2019.06.002.

Tilloy et al. (2019) A review of quantification methodologies for multi-hazard interrelationships, Earth-Science Reviews, Volume 196, 102881, https://doi.org/10.1016/j.earscirev.2019.102881.

- P2:L17: I uspect the official death toll be close to 600 (https://www.unocha.org/southern-and-eastern-africa-rosea/cyclones-idai-and-kenneth). Please double check.

- P3:L15-19: A useful citation: Sadegh et al. (2018) Multihazard scenarios for analysis

of compound extreme events, Geophysical Research Letters 45 (11), 5470-5480, doi: 10.1029/2018GL077317.

- P10:L1: Please, clearly explain how you made this conclusion " This assumption seems reasonable based on visual observations..."

- P10:L3: "If no co-occurrences were measured (X=0), we select Pc = p", while from equation 5, if X -> 0 then Pc -> 0. There is mathematical inconsistency here.

- P10:L14: Not sure if a 5-yr event fits in the definition of compound "extremes" that has been used in the title.

- P10:L15: Many previous studies have found Archimidean copulas preferable in joint extreme analysis; and as you correctly mention in page 16, appropriate characterization of correlation structure can significantly affect the estimation of return period of compound extremes. Justify, why Gaussian Copulas used here?

Nice job!

---

## Author Comment (AC1) · 25 Oct 2019

*We would like to thank the reviewer for taking the time to critically review our manuscript. In this document, we respond to the comments received point by point and show the changes suggested in the manuscript by referring to the page and line numbers in the revised manuscript. Our responses are shown in blue and in italics.*

General Comments:

Couasnon et al., present a global-scale analysis of compound flood hazard potential. Using modeled datasets of river discharge and storm surge, they highlight locations where the potential for compound flooding may have been overlooked using observational records alone, thus extending their analysis beyond that of Ward et al., 2018. They also incorporate new, creative metrics beyond conditional dependencies and copulas to describe the likelihood for co-occurrence of these variables. I feel that the research is an important contribution to the literature as it provides a global perspective on where compound flooding may matter for flood risk assessment and more detailed/localized studies. The manuscript is well written and describes the motivation, methods, and results very clearly, making it an overall pleasure to read. I have a few minor comments in which I believe the authors would benefit from addressing.

*Many thanks for these kind words, we are pleased that the reviewer finds the manuscript to be interesting, relevant and well written.*

Specific Comments:

I appreciate that the authors consistently describe their results as "flood potential." Often the literature characterizes an extreme forcing (e.g., storm surge) as an extreme flood, without ever linking it to impacts. The authors make clear they are describing how the compounding forcing has the potential to drive flooding. On that note, I think that the title is slightly misleading/redundant, as there are really no implications for flood hazard and/or I'm not sure how that's different from "measuring compound flood potential." There are a few other locations in the text (noted in my comments below) that the authors could add "potential" to as well.

*Thanks for the suggestions: we have removed the last part of the initial title in the revised manuscript:*

*[page 1, L1-2]"Measuring compound flood potential from river discharge and storm surge extremes at the global scale".*

*We have also added the word "potential" at the places indicated, as described later in this document.*

5   The majority of Central America, South America, and Southern Africa have a poor hit rate for storm surge (not quite as bad for discharge). Meanwhile, some of these locations have a very high number of modeled co-occurring annual maxima and correlation coefficients for (Sn,qn) (e.g., Chile and South Africa). In Section 3.4, the authors do discuss some of the limitations of being able to represent the accuracy of the model in these locations, and also warn that the results of low compound flood potential in the northern latitudes is uncertain and should be interpreted with care. Should a similar warning

10   be provided in other locations? Do the authors have any indication they may be overestimating flood hazard potential in areas where the models do not accurately depict observations, yet have a high amount of co-occurring events?

*Even though we used state-of-the-art global models for both the discharge and storm surge variables, we are aware that their performance varies spatially due to various reasons as reported in our original manuscript. We complemented*

15   *performance analysis presented in the original studies of Beck et al.(2017); Muis et al. (2016) and Schellekens et al. (2017) by specifically including relevant metrics for our study, i.e. the timing of extremes and the ranking of extremes, with observations. However, as pointed by the reviewer, we did not directly compare the results from our study with those based on observations. This has now been added in the revised manuscript in an additional Supplement (section S1.3). Due to the selection criteria we impose, this resulted in just 25 pairs of observation stations, mainly located in Europe. Therefore, the*

20   *insight we gained remains limited to very specific locations and does not allow for a comparison at larger spatial scales. In general, we find that using the simulated discharge and storm surge variables captures the sign of the dependence correctly, but that the magnitude of this dependence can largely vary. Moreover, the models tend to correctly identify the locations with the highest number of co-occurring discharge and storm surge annual maxima but overestimate this number. This emphasizes the need to perform in-depth local studies that should include local data if available and/or calibrated models*

25   *when quantifying local compound flood risk.*

*In the revised manuscript, , we modified the text to highlight these points:*

- *In the methods section 2.1:*

  *[page 6, L4-12]: "We further assess how the respective performance of both models can affect the compound flood potential measures defined in section 2.3 (see Supplement S1.3). Due to the selection*

30   *criteria we impose, this results in 25 pairs of observation stations, which is insufficient for a rigorous comparison at large regional scales. In general, we find that using the simulated discharge and storm surge variables captures the sign of the dependence correctly but the magnitude of this dependence can largely vary. Moreover, the models tend to correctly identify the locations with the highest number of co-occurring discharge and storm surge annual maxima but overestimate this number. This additional*

*validation showed that the performance of both models vary globally which as a result can locally bias the compound flood potential. Nevertheless, it provides an acceptable performance on average for the purpose of this study, i.e. to provide a first-cut assessment of the compound flood potential at the global scale."*

5      •    *In the discussion section 3.4:*

*[page 19, L4-11]: "Moreover, the results presented in this study are dependent on the accuracy of the models. The validation performed in this study indicates an acceptable performance on average, albeit with large spatial differences. The timing of the simulated storm surge compared with observations shows a poor performance of the model for stations along the coasts of South America and Africa. Yet, the*

10      *accuracy of the models in measuring the compound flood potential along large parts of the coastline in South America, Africa, and Asia could not be assessed due to a lack of long-term gauge observations of discharge and sea levels. This was the main motivation for examining different compound flood measures and selecting moderate joint return period conditions, in order to identify regions potentially exposed to this phenomenon."*

15      *[page 19, L17-18]: "More generally, this emphasizes the need for local-scale studies in order to accurately quantify compound flood hazard locally."*

On that note, on Page 5-6: Line 34 – Line 1, the authors state, "As a result, the timing and correlation of extreme storm surge is generally well represented . . . and less well captured for the South African and South American coasts." This statement

20   seems to be putting it lightly, as the hit rate is 0 for all these locations, and the correlation coefficient is negative, or relatively low. It more or less seems that it doesn't capture the South African and South American coasts.

*We agree with the reviewer, this was an oversight from our side. We have modified the text in the revised manuscript to:*

*[page 6, L3-4]" As a result, the timing and correlation of extreme storm surge is generally well represented along the European, North American, Japanese, and Australian coast, but not for the South*

25      *African and South American coasts."*

*[page 19, L4-7]: "The validation performed in this study indicated an acceptable performance on average, albeit with large spatial differences. The timing of the simulated storm surge compared with observations shows a poor performance of the model for stations along the coast of South America and Africa."*

30   Finally, the authors state in the conclusions (Page 18, Line 8) that, "Our study provides a first indication of locations where discharge and storm surge interactions are strong." Technically, the authors are not investigating "interactions" between these variables. Furthermore, Ward et al., 2018 provided the first indication of locations where these dependencies between variables are strong.

*We thank the reviewer for noting this. We have modified this sentence in the revised manuscript to:*

> *[page 20, L10-12]: "This is also the case for current global state-of-the-art flood models, and our study provides a first indication of regions along the global coastline where discharge and storm surge extremes are likely to co-occur."*

Comments on Specific Lines:

Page 5, Line 30: Hit rate is defined in the manuscript, but not here. Perhaps the authors could add it to the preceding sentence, "We calculate the percentage of annual maxima dates correctly predicted, termed hit rate . . ."

*We agree and thank the reviewer for this suggestion, the text has been modified to:*

> *[page 5, L30-31]" We calculate the percentage of annual maxima dates correctly predicted, termed "hit rate", and the Spearman's rank correlation coefficient..."*

Page 6, Line 28: "We transform the annual maxima pairs to probability space using their respective empirical cumulative distribution functions" – I'm confused as to how this is done for 3 points (in this example at least). Am I missing something

15   here?

*Our apologies for the lack of explanation, the use of the word "their" in our original sentence was confusing. The transformation to probability space is done using the marginal empirical cumulative distribution functions of the discharge $Q$ and the storm surge annual maxima, $S$. But we show the co-occurring events in red and denote them as $(Q*n, S*n)$. In the revised text, we clarified this sentence:*

> *[page 7, L10-13]" We transform the annual maxima pairs $(Q_n, S_n)$ to probability space using the empirical cumulative distribution functions of both variables (Figure 1f). The pseudo-observations of the co-occurring events (shown in red) do not only correspond to joint high quantiles but also a combination of high, moderate, and low storm surge with moderate to high quantiles of discharge.*

25   Page 8, Line 19: I think simulated is sufficient when describing the datasets, and then can be removed from the rest of the text describing the datasets. The readers will know they're not observations. Anywhere else in the text makes me think there was a statistical simulation instead. "Simulated" is also used on Page 13, Line 10, caption Figure 5, and in a few other locations in the manuscript.

*We agree and we thank the reviewer for this suggestion. We have removed the word "simulated" in the rest of the text after*

30   *section 2.1.*

Page 10, Line 8: Similar to the above comment, how are the authors finding the dependence of (Q*, S*) pairs, when there may only be 1 or 2?

*Our apologies for the lack of explanation. If there are 0 or 1 pairs in the sets of* (Q\*, S\*) *pairs, we cannot calculate the correlation and therefore use the dependence between (Q,S) pairs. If there are 2* (Q\*, S\*) *pairs, we use the described bootstrapping method but we already know that the result will not be statistically significant since the correlation obtained will be 1 or -1. In the revised text, we clarified the sentence:*

5            [page 11, L4-5]*"Note that the latter is possible only if we observe at least two pairs of ( Q\*, S\*) pairs."*

Page 17, Line 20: The authors state that some approaches, "may result in sampling events that are not extremes and underestimating the compound flood potential measure." At the same time, the definition of a compound event also includes events driven by extreme and non-extreme variables, so the authors could be underestimating the types of events that could

10   drive extreme flooding.

*We agree with the reviewer that compound flood events can include combinations of extreme driver variables with non-extremes ones. However, finding these combinations requires starting from the impact variable and looking at the corresponding values of the drivers variable, i.e. a bottom-up approach. In our study this is not feasible since we do not force these drivers to obtain an impact variable (for example, the flood depth or the flood extent). Therefore, within our*

15   *adopted methodology, using a lower threshold would mainly add a lot of noise into the compound flood potential metrics we are measuring ( the correlation coefficient and number of co-occurrences) as we cannot clearly discriminate between what causes impact from what does not.*

*We modified the text to better reflect this point:*

        [page 18, L20-21]*:" For example, they may result in sampling events that are not relevant for the*

20        *flood hazard analysis and therefore add some noise in the sets of events used to measure the*

        *compound flood potential."*

Page 18, Line 26: "developed compound flood potential indicators" This is the first time the authors use the word indicators, so I'm not sure which metrics they're referring to. I suggest removing this, or defining it earlier on.

25   *We agree with the reviewer and have removed this part:*

        [page 19, L30]*:" We analysed important characteristics of compound flooding related to the timing*

        *and joint dependence between river discharge and storm surge extremes."*

Figures and Tables: Figure 1: What is (Qn, Sn) on plot 1e? This hasn't seemed to be defined. Furthermore, is there a reason

30   the authors choose to use Mexico (Section 2.2) as the example when Madagascar is the chosen example location later?

*Our apologies for the lack of explanation. (Qn, Sn) have been defined in text on page 7 but indeed this is not obvious when reading the caption of Figure 1.There was no specific reason for selecting Mexico other than to show a location not covered*

*by global observation datasets. As suggested by the reviewer, we have modified the example for the location on Madagascar and modified the title of Figure 1:*

> [page 8, L1-5]:*" Figure 1: Maximum daily storm surge, $s$, (a) and daily discharge , $q$, (b) paired for a*

5 *location along the coast of Madagascar (c). Discharge Q and storm surge S annual maxima from both (a) and (b) are used to construct event time-series based on the conditional sampling method, $(Q_n, s_n)$ and $(S_n, q_n)$, using a time window of $\Delta = 3$ days(d) and joint annual maxima $(Q_n, S_n)$ (e). Joint annual maxima co-occurring within $\Delta = 3$ days, $(Q_n^*, S_n^*)$ , are shown in red. Events shown in (e) are shown in probability space in (f).*

10 Figure 2b: The authors may want to consider adding a zoomed inset to show what is described in the text in Line 9-11 on this page.

*We thank the reviewer for this suggestion. We think however that adding a zoomed inset in Figure 2a would still be difficult to read if using the same scale for the density (y-axis). This was the major reason for showing Figure 2b, which shows a*
15 *similar information as Figure 2a but as a CDF and using a logarithmic scale. In Figure 2b, the shape of the curves cannot be seen for X < than 5 days, whereas this is not the case for Figure 2a. We highlighted the complementarity of the figures in the revised text:*

- *[page 9, L13-14]:" Figure 2: Probability mass function for the probability of observing X co-occurrences of annual maxima in N=35 years and for $\Delta = 3$ days (a). Same as (a) but shown as the*
20 *exceedance probability $P(X \geq x)$ (b). Note that the y-axis for (b) is logarithmic."*
- *[page 9, L18 and page 10, L1]: ", see Figure 2b. As the number of co-occurrences increases, the exceedance probability of observing…"*

Figure 5: I'm a bit confused about what the black line displays. Is it the empirical distribution of all AM occurrences across
25 the globe? If so, how is this used to describe dependence at a particular location? I assumed that the red/black line could be applied to any location, e.g., if these two variables were independent over these time scales, here's the probability it would occur. But if the black line is generated for all locations, I don't understand how it can be applied to a specific location. Am I misinterpreting this?

30 *As correctly noted by the reviewer, in Figure 5 we show results globally, i.e. across all locations whereas in the Method section we focus on a given location. This is indeed confusing and in the revised text, we clarify and provide further explanations in the Methods section:*
- *[page 9, L4-5]:" We analyse the number of co-occurrences of annual maxima of river discharge and storm surge along the global coastline."*

- [page 10, L3-11]:*" For our example location, we observe $x = 14$ co-occurrences. Assuming that both annual maxima occur randomly within the year, we read from Figure 2b an exceedance probability of $P(X \geq 14) \approx 1.10^{-15}$ , whereas if they can occur randomly within the same season of 90 days this probability increases to $P(X \geq 14) \approx 1.10^{-7}$, but in both cases remain a very low probability. The right tail of the distributions in Figure 2b nevertheless show that one could expect to observe some locations with a large number of co-occurrences even under independence, but these situations are expected up to a certain frequency. In other words, if the total number of co-occurrences along all stations would follow statistically independence, we would expect to observe this situation at none of the stations (=.$P(X \geq 14) \times 3,434$ stations). Therefore, we compare the distributions of co-occurrences along the global coastline obtained with the ones shown in Figure 2."*

Supplemental Table S3: Do these numbers denote averages across all locations?

*These numbers denote the fraction of locations with a positive statistical dependence for different confidence intervals. In the revised text, we modified the title of the Table and indicate the unit in the title to make this point clearer:*

> [Supplement S3 page 9]*" Table S1: Total percentage (%) of paired locations along the global coastline with a positive and statistically significant Spearman's rank correlation coefficient both for $(Q_n, s_n)$ and $(S_n, q_n)$ pairs and significance levels of $\alpha = 0.05$ and $\alpha = 0.10$"*

Technical Corrections: Page 6, Line 5: Better stated as, "This leads to 3,979 stations of paired river discharge and storm surge. . ."

*We modified the text as suggested by the reviewer:*

> [page 6, L16-17]: *"This leads to 3,434 stations of paired river discharge and storm surge time-series between 1980 to 2014, representing 35 years of daily data." ***

*\*\*Note that since the original manuscript was submitted, we discarded some stations because their upstream catchment area was smaller than 1,000 $km^2$. All the figures and results in the manuscript have been modified to reflect this change. This does not change any of the conclusions from our study.*

Page 8, Line 2: Compound Flood potential "from extremes" (since you're not including non-extreme forcing)

*For this specific sentence, we think that adding "from extremes" might be confusing as we are already referring to the "sets of events". The way we construct these sets of events results in one variable being extreme while the other might not be, as illustrated in Figure 1d of the original manuscript. Both sets of events resulting from the conditional sampling method combine the annual maxima of one time-series with values of the other series that are not necessarily extremes. Therefore in*

*this particular case, we think that adding "from extremes" might be confusing. We agree that this sentence should be clarified and suggest the following:*

> *[page 8, L8-9]: "We use the different sets of events constructed from the marginal extremes as defined in Section 2.2 to measure the compound flood potential at all the paired locations in three ways."*

Page 9, title 2.2.3 "Quantification of the Compound Flood Hazard Potential" (add potential)

*We modified the text as suggested by the reviewer:*

> *[page 10, L12]: "Quantification of the Compound Flood Hazard Potential"*

Page 17, Line 6: Add "potential" e.g., . . ."we identify potential compound flooding hotspots. . ."

*We modified the text as suggested by the reviewer:*

> *[page 18, L6]: "While we identify potential compound flooding hotspot regions from extreme discharge and storm surge variables, substantial uncertainties remain as to how this phenomenon will propagate into inland flooding."*

If the citation Eilander et al,. 2019 has been updated to a more final format, please update citation.

*We carefully reviewed all the references in our text and updated the ones that are now in a final format, i.e.:*

[revised manuscript text omitted]

**Supplement S1. Validation of annual maxima daily discharge and storm surge**

For this study we select the simulated  runoff from the JULES model,  routed with the CaMa-Flood model based on the performance tests presented in Beck et al. (2017) and Schellekens et al. (2017). Here, we complement these tests by looking at the rank correlation coefficient and the absolute average lag in the timing of the annual maxima in
5 observations with a record length of at least 20 years. We also apply similar performance tests for the storm surge variable.

We use the Spearman's rank correlation, a nonparametric measure for monotonic relationships between two variables. The rank correlation coefficient is equivalent to the Pearson's product moment correlation, $\rho$, applied to the ranks of the annual maxima both observed ($X_o$) and simulated ($X_s$), such that:

$$r_{X_o,X_s} = \rho(X_o, X_s)$$

We also calculate a simple metric often used in flood forecasting studies, the Hit Rate, $H$, but applied to the date of the
10 annual maximum. This corresponds to the probability of detection of the date of the annual maxima. We assume that the simulated date of the annual maximum $D_S^i$ is correctly represented if it is within $\pm$ 3 days of the observed annual maximum $D_O^i$ in the $i$-th year considered:

$$H = \frac{\sum_{i=1}^{N}(D_S^i \cap D_O^i)}{N}$$

**S1.1 Annual maxima of daily discharge**

We compare the performance of the modelled annual maxima of daily discharge with discharge observations from the
15 Global Runoff Data Base (GRDB) from the Global Runoff Data Centre[1] . We follow a similar procedure as described in Zhao et al. (2017) to select stations in near-natural areas, and therefore minimise anthropogenic influence on the measured discharge. A catchment is selected if less than 2% of its upstream area is subject to irrigation, if the total reservoir capacity in the catchment is less than 10% of its long-term mean annual discharge, if its catchment area is at least 1000 km$^2$ or higher, and if the record length is at least 20 years with a minimum completeness of 75% per year within the period 1980-2014. This
20 leads to the selection of  1116 stations, shown in Figure S1 and S2. The timing of the simulated discharge annual maxima compared with observations varies greatly globally. We find a median hit rate of 0.21 (min:0, max:0.79, s.d.:0.18) and a median rank correlation coefficient is 0. 57 (min: -0.35, max: 0.96, s.d.: 0. 22).
* * *
[1] The Global Runoff Data Centre, 56068 Koblenz, Germany www.bafg.de/GRDC/EN/Home/homepage_node.html

[Figure]

**Figure S1: Probability of correctly detecting the date of the discharge annual maxima within ± 3 days.**

[Figure]

**Figure S2: Spearman's rank correlation between the  discharge annual maxima  obtained from the model and from the observations.**

**S1.2 Annual maxima of storm surge**

In order to compare the simulated storm surge variable with observations, we extract the equivalent of the storm surge from the sea levels observations of the Global Extreme Sea-level Analysis Version 2 database (GESLA-2) database (Woodworth et al., 2017). We select coastal stations if they have at least 20 years of data and a minimum completeness of 75% per year and compare it with the closest GTSM output location within a maximum radius of 20 km. This leads to the selection of 165

stations, shown in Figure S3 and S4. The timing of the simulated storm surge annual maxima compared with observations varies greatly globally. We find a median hit rate of 0.342 (min: 0, max: 0.70, s.d.: 0.221) and a median rank correlation coefficient of 0.37 (min: -0.45, max: 0.81, s.d.:0.31).

[Figure]

5    **Figure S3: Probability of correctly detecting the date of the storm surge annual maxima.**

[Figure]

**Figure S4: Spearman's rank correlation between the storm surge annual maxima occurrence obtained from the model and from the observations.**

**S1.3 Compound flood potential measure from simulated and observed time series**

In order to assess how the performance of the global models presented in section S1.1 and S1.2 might affect the results presented in this study, we compare the covariability of the discharge and storm surge. To do so, we start with the combinations of discharge and tide stations presented in Ward et al. (2018) and filter this list to keep combinations that have at least 75% of overlapping data per year and a minimum of 20 years within the period 1980-2014. We only keep GESLA-2 tidal stations for which a GTSM output location is within a 20 km radius from the latter and GRDB discharge stations in near-natural areas as described in Zhao et al. (2017). This leads to the selection of 25 combinations of stations with the majority being located in Europe. Figure S5 presents the conditional dependence strength (Fig. S5a and S5b) and the total number of co-occurring annual maxima (Fig. S5c) obtained from the paired observations stations and the corresponding model output locations.

[Figure]

**Figure S5: Comparison of the spearman's rank correlation between the storm surge conditional on annual maxima discharge (a), the discharge conditional on annual maxima storm surge (b) and the number of co-occurring annual maxima within a 3-day time window (c) obtained from the simulated discharge and storm surge variables (x-axis) and from observations (y-axis) .**

Concerning the conditional dependence, we observe an overall positive agreement but with a large spread between the model outputs and the observations. This is particularly noticeable in Figures S6 and S7, which show the corresponding results at the paired stations. Some locations, such as in the south of England or the southwest of Australia, exhibit a similar dependence behavior as with the observations, albeit with some bias, while other locations, such as in Italy or the northwest coast of Australia, show opposite results. Nevertheless, we note that at most locations the model can capture the sign of the correlation correctly. Figure S8 indicates that, in general, the selected models tend to correctly identify the locations with the highest co-occurrences but overestimate the number of co-occurring annual maxima. An outlier is the northwest of Australia, where no co-occurring annual maxima are measured based on observations and 10 based on the simulated variables. These discrepancies could be due to the fact that the selected global models fail to capture important small-scale features and processes driving extreme discharge and storm surge at these locations.

[Figure]

(a) From observation stations

[Figure]

(b) From the simulated variables

Figure S6: Spearman's rank correlation between the storm surge conditional on annual maxima discharge obtained from observation stations (a), and from the simulated variables at these locations (b) .

[Figure]

(a) From observation stations

[Figure]

(b) From the simulated variables

Figure S7: Spearman's rank correlation between discharge conditional on annual maxima storm surge obtained from observation stations (a), and from the simulated variables at these locations (b) .

[Figure]

**(a) From observation stations**

[Figure]

**(b) From the simulated variables**

**Figure S8: Number of co-occuring annual maxima of discharge and storm surge from observation stations (a), and from the simulated variables (b) .**

**Supplement S2. Co-occurrences of joint annual maxima at selected locations**

[Figure]

Figure S5S9: Examples of pseudo-observations from simulated annual maxima of discharge Q and storm surge S at selected locations. Red dots indicate a co-occurrence of Q and S, $(Q^*, S^*)$, within a time lag of 3 days.

**Supplement S3. Sensitivity of time window on Spearman's $r_s$ correlation coefficient**

| Time window Δ (days) | Storm surge conditional on discharge annual maxima $(Q_n, s_n)$ | | Discharge conditional on storm surge annual maxima $(S_n, q_n)$ | |
|---|---|---|---|---|
| | α = 0.05 | α = 0.10 | α = 0.05 | α = 0.10 |
| 0 | 11 | 7 | 8 | 4 |
| 1 | 144 | 0.19 | 0.19 | 0.24 |
| 2 | 0.14 | 20 | 0.19 | 4 |
| 3 | 0.14 | 20 | 19 | 0.25 |
| 4 | 0.15 | 20 | 8 | 0.24 |
| 5 | 0.15 | 21 | 8 | 0.24 |
| 6 | 0.15 | 0.21 | 9 | 0.24 |
| 7 | 0.16 | 22 | 0.18 | 0.24 |

**Table S1:**  Total percentage (%) of paired locations along the global coastline with a positive and statistically significant Spearman's rank correlation coefficient both for $(Q_n, s_n)$ and $(S_n, q_n)$ pairs and significance levels of α = 0.05 and α = 0.10.

[Figure]

**Figure S6S10**: Spearman's $r_s$ correlation coefficient between storm surge conditional on discharge annual maxima ($Q_n$, $s_n$) for a time window of $\Delta = 0$ days (top) and $\Delta = 7$ days (bottom). Black dots denote locations with no significant dependence ($\alpha = 0.05$).

[Figure]

**Figure S7S11**: Spearman's $r_s$ correlation coefficient between storm surge conditional on discharge annual maxima $(Q_n, s_n)$ for a time window of $\Delta = 0$ days (top) and $\Delta = 7$ days (bottom). Black dots denote locations with no significant dependence ($\alpha = 0.10$).

[Figure]

Figure S8S12: Spearman's $r_s$ correlation coefficient between discharge conditional on storm surge annual maxima ($S_n$, $q_n$) for a time window of $\Delta = 0$ days (top) and $\Delta = 7$ days (bottom). Black dots denote locations with no significant dependence ($\alpha = 0.05$).

[Figure]

Figure S913: Spearman's $r_s$ correlation coefficient between discharge conditional on storm surge annual maxima $(S_n, q_n)$ for a time window of $\Delta = 0$ days (top) and $\Delta = 7$ days (bottom). Black dots denote locations with no significant dependence ($\alpha = 0.10$).

**Supplement S4. Influence of the time window on _the_ number of co-occurring annual maxima**

[Figure]

Figure S14: Number of co-occurring yearly maxima of storm surge and discharge obtained between 1980-2014 using a time window of $\Delta = 0$ days (top) and $\Delta = 7$ days (bottom).

---

## Author Comment (AC2) · 25 Oct 2019

5 *We would like to thank the reviewer for taking the time to critically review our manuscript. In this document, we respond to the comments received point by point and show the changes suggested in the manuscript by referring to the page and line numbers in the revised manuscript. Our responses are shown in blue and in italics.*

General Comments:

10 Authors explore the potential of compound flooding due to river flow and storm surge along the coast lines globally. They use numerical simulations forced by reanalysis dataset to extend our understanding about this phenomenon beyond previously reported regions with in-situ observational data. Characterization of compound flooding hazards is a very important problem in coastal regions worldwide and helps improving hazard prediction and effective resource allocation for flood risk management. The idea is interesting, study is robustly designed and manuscript is very well written. Given the fact

15 that this is a significant contribution to the community of coastal hazard and it could attain the readership of a broader community of natural hazard researchers, I recommend it for publication in NHESS after a minor revision. I am mainly concerned about the significance of conclusions made here, compared to previously reported patterns and results. Below, I provide more detailed comments and suggestions:

20 *We are very pleased that the reviewer finds the article to be valuable and well written and we thank the reviewer for the encouraging comment.*

- In the abstract you mention ". We find many hotspot regions of compound flooding that could not be identified in previous global studies based on observations alone, such as: Madagascar..." and then further explore Madagascar as a case study. My

25 question is that, given the fact that there is no observational record of discharge and storm surge in or at close proximity of Madagascar (Figures S1-S4), how reliable such compound hazard hotspot detection would be (seems among the hottest)? In other words, while with such limited information, estimation of individual extremes will be associated with significant uncertainty and errors associated with capturing the timing of extremes will add-up (Page 5), how conclusive your pattern detection would be? And why did not choose another location with more reliable record? I see you have thoroughly

30 discussed the limitation of this work in pages 17 and 18, but still the audience needs to know the significance of results in Figure 3.

*The reviewer raises very valid points. The main objective of our work is to provide an analysis along the global coastline of the compound flood potential from river discharge and storm surge. We believe that this is especially relevant for locations without any observations, since these locations could not be included in studies based on observations such as Ward et al. (2018). However for Madagascar, since we do not have observational records to compare our results with, we cannot calculate how reliable those estimates are. Even though we used state-of-the-art global models, given the performance of the models and the limited length of the time series (35 years), we also do not aim to provide underline{precise} estimates about the compound flood hazard potential as we think that this would be misleading. Instead, our approach focusses on extracting as much information as possible from the modelled data by analysing underline{different} sets of events in order to measure the compound flood potential. Regions where we consistently observe some high compound flood potential from the different sets of events will result with a high compound flood hazard potential. We also critically reflect on our results by comparing such patterns to existing scientific literature. With this in mind, we used the term "hotspot location" to refer to locations where we consistently observe some compound flood potential that deviates from statistical independence We agree with the reviewer that this term may be misleading without proper explanation and have therefore removed it from the abstract.*

*As rightly pointed out by the reviewer, the uncertainty and errors in the modelling of river discharge and storm surge will add-up. This means that, in regions where the models do not perform well, the compound flood potential can be underestimated or overestimated. In the revised manuscript, we added an additional Supplement section (Section S1.3) to compare the results from our study with those based on observations. Due to the selection criteria we impose, this resulted in 25 pairs of observation stations, mainly located in Europe. This additional analysis does not show a clear over- or underestimation and the sparse observations do not allow for a rigorous comparison at regional scales. It clearly highlights, however, the need for local-scale studies in order to properly quantify compound flood hazard locally.*

*In the revised manuscript, we modified the text to highlight these points:*

- *In the abstract:*

  *[page 1, L24-25]" The purpose of this study is to fill this gap and identify regions with a high compound flooding potential from river discharge and storm surge extremes in river mouths globally."*
  *[page 1, L28-29]" Our analysis indicates many regions that deviate from statistical independence and could not be identified in previous global studies based on observations alone…".*

- *In the methods section 2.1:*

  *[page 6, L4-12]: "We further assess how the respective performance of both models can affect the compound flood potential measures defined in section 2.3 (see Supplement S1.3). Due to the selection criteria we impose, this results in 25 pairs of observation stations, which is insufficient for a rigorous comparison at large regional scales. In general, we find that using the simulated discharge and storm surge variables captures the sign of the dependence correctly but the magnitude of this dependence can*

*largely vary. Moreover, the models tend to correctly identify the locations with the highest number of co-occurring discharge and storm surge annual maxima but overestimate this number. This additional validation showed that the performance of both models vary globally which as a result can locally bias the compound flood potential. Nevertheless, it provides an acceptable performance on average for the*

5    *purpose of this study, i.e. to provide a first-cut assessment of the compound flood potential at the global scale."*

- *In the discussion section 3.4:*

   [page 19, L4-11]: *"Moreover, the results presented in this study are dependent on the accuracy of the models. The validation performed in this study indicates an acceptable performance on average, albeit*

10   *with large spatial differences. The timing of the simulated storm surge compared with observations shows a poor performance of the model for stations along the coasts of South America and Africa. Yet, the accuracy of the models in measuring the compound flood potential along large parts of the coastline in South America, Africa, and Asia could not be assessed due to a lack of long-term gauge observations of discharge and sea levels. This was the main motivation for examining different compound flood measures*

15   *and selecting moderate joint return period conditions, in order to identify regions potentially exposed to this phenomenon."*

   [page 19, L17-18]: *"More generally, this emphasizes the need for local-scale studies in order to accurately quantify compound flood hazard locally."*

20   - Useful citation for Introduction Santiago-Collazo et al. (2019) A comprehensive review of compound inundation models in low-gradient coastal watersheds, Environmental Modelling & Software, Volume 119, Pages 166-181, https://doi.org/10.1016/j.envsoft.2019.06.002.

   Tilloy et al. (2019) A review of quantification methodologies for multihazard interrelationships, Earth-Science Reviews, Volume 196, 102881, https://doi.org/10.1016/j.earscirev.2019.102881.

25   *Many thanks for these suggestions. The revised manuscript now includes the references:*

   [page 2, L29-30]: *"Yet, these interactions can significantly influence the magnitude of simulated water levels (Santiago-Collazo et al., 2019)"*

   [page 3, L13-14]: *"A consistent mathematical definition of compound flood events does not exist and multiple statistical methods have been suggested to study this phenomenon (Hao et al., 2018; Tilloy et al.,*

30   *2019)"*

   - P2:L17: I uspect the official death toll be close to 600 (https://www.unocha.org/southern-and-eastern-africa-rosea/cyclones-idai-andkenneth). Please double check.

*As suggested, we double-checked the values mentioned in the introduction. The link mentioned by the reviewer indeed mentions an official death toll of 602 casualties but this value is only taking into account Mozambique. When including Zimbabwe and Malawi, the reported total number of casualties increases to 960. At the time of writing, it was difficult to find final estimates concerning the casualties due to Cyclone Idai. We have now modified our text to include this piece of information:*

[page 2, L16-20]*:" It was reported that Idai directly affected 3,000,000 people, caused at least 960 casualties, destroyed about US$1 billion in infrastructure, ruined 500,000 hectares of crops, and caused widespread power outages, and multiple road closures that complicated aid distribution and the humanitarian interventions to keep cholera outbreaks under control (Bloomberg, 2019; ERCC, 2019; USAID, 2019)"*

- P10:L1: Please, clearly explain how you made this conclusion " This assumption seems reasonable based on visual observations..."

*Our apologies for this lack of explanation. We have now added some further clarifications in the text:*

[page 10, L17-21]*:" Here, we assume that $P_c$ is not a function of the threshold considered. This assumption seems reasonable based on visual observations at randomly selected locations (see Supplement S2) since the pseudo-observations of co-occurring annual maxima (red circles in Figure S9) are not concentrated in a specific area of the probability space . Therefore, at a given paired location, we approximate $P_c$ with the following equation:. "*

- P10:L3: "If no co-occurrences were measured (X=0), we select Pc = p", while from equation 5, if X -> 0 then Pc -> 0. There is mathematical inconsistency here.

*Apologies for this lack of explanation. In the case where no co-occurrences were measured, using equation 5 would lead to $P_c = 0$ and thus $P_F = 0$, implying that there is no compound flood hazard potential. However, from section 2.3.2, we cannot rule out that co-occurrences will never occur at a given location since we know that even under complete statistical independence, there is a chance for these two variables to co-occur. Therefore in order to measure the compound flood hazard at locations where no co-occurrences were observed, we select $P_c = p$. Here, we selected the probability p assuming that both variables can randomly occur within the year.*

*We modified the text to clarify this:*

*[page 10, L20-23]:" Therefore, at a given paired location, we approximate $P_c$ with the following equation:*

$$\begin{cases} P_c = p \, , & X = 0 \\ P_c = \dfrac{X}{N} \, , & X > 0 \end{cases} \tag{5}$$

*If no co-occurrences were measured ($X = 0$), we cannot rule out that co-occurrences will never happen and we select $P_c = p$, i.e. the probability of observing a co-occurrence under independence in any given year."*

- P10:L14: Not sure if a 5-yr event fits in the definition of compound "extremes" that has been used in the title.

*In the manuscript, the term 'extremes' refers to the subset of the extreme value population obtained by sampling the block maxima series. While our methodology can be applied for higher return periods, the main motivation for selecting the 5-year event magnitude is due to the limited sample size, the uncertainties in the simulated variables at some location and the selection of the Gaussian copula.*

*We would also like to point out that in areas without specific flood protection strategies, a 5-year discharge event may be enough to cause impactful flooding. This is for example particularly relevant in countries with no or low flood protection standards, as reported in Scussolini et al. (2016). Overbank flooding from unprotected rivers can already happen for return periods higher than 1.5 years (Leopold, 1978) and for this reason is a common assumption in global river flood models (Ward et al., 2013).*

*We modified the text to clarify this:*

*[page 11, L7-10]:" Given the limited temporal coverage of the data, we present the result for a quantile threshold equivalent to a 5-year discharge magnitude and a 5-year storm surge magnitude (i.e., $u = v = 0.8$). While this represents a relatively low threshold value, we note that such conditions can be sufficient to cause flooding in areas with no or low flood protection standards (see Scussolini et al. (2016) for a global overview). Overbank flooding from unprotected rivers can already happen for discharge return periods higher than 1.5 years (Leopold, 1978) and result in damaging floods when impacting human livelihoods (Ward et al., 2013)."*

[Figure]

**Figure S1: Probability of correctly detecting the date of the discharge annual maxima within ± 3 days.**

[Figure]

**Figure S2: Spearman's rank correlation between the  discharge annual maxima  obtained from the model and from the observations.**

**S1.2 Annual maxima of storm surge**

In order to compare the simulated storm surge variable with observations, we extract the equivalent of the storm surge from the sea levels observations of the Global Extreme Sea-level Analysis Version 2 database (GESLA-2) database (Woodworth et al., 2017). We select coastal stations if they have at least 20 years of data and a minimum completeness of 75% per year and compare it with the closest GTSM output location within a maximum radius of 20 km. This leads to the selection of 165

stations, shown in Figure S3 and S4. The timing of the simulated storm surge annual maxima compared with observations varies greatly globally. We find a median hit rate of 0.32 (min: 0, max: 0.70, s.d.: 0.21)  and a median rank correlation coefficient of  0.37 (min: -0.45, max: 0.81, s.d.:0.31).

[Figure]

5    **Figure S3: Probability of correctly detecting the date of the storm surge annual maxima.**

[Figure]

**Figure S4: Spearman's rank correlation between the storm surge annual maxima  obtained from the model and from the observations.**

**S1.3 Compound flood potential measure from simulated and observed time series**

In order to assess how the performance of the global models presented in section S1.1 and S1.2 might affect the results presented in this study, we compare the covariability of the discharge and storm surge. To do so, we start with the combinations of discharge and tide stations presented in Ward et al. (2018) and filter this list to keep combinations that have at least 75% of overlapping data per year and a minimum of 20 years within the period 1980-2014. We only keep GESLA-2 tidal stations for which a GTSM output location is within a 20 km radius from the latter and GRDB discharge stations in near-natural areas as described in Zhao et al. (2017). This leads to the selection of 25 combinations of stations with the majority being located in Europe. Figure S5 presents the conditional dependence strength (Fig. S5a and S5b) and the total number of co-occurring annual maxima (Fig. S5c) obtained from the paired observations stations and the corresponding model output locations.

[Figure]

**Figure S5: Comparison of the spearman's rank correlation between the storm surge conditional on annual maxima discharge (a), the discharge conditional on annual maxima storm surge (b) and the number of co-occurring annual maxima within a 3-day time window (c) obtained from the simulated discharge and storm surge variables (x-axis) and from observations (y-axis) .**

Concerning the conditional dependence, we observe an overall positive agreement but with a large spread between the model outputs and the observations. This is particularly noticeable in Figures S6 and S7, which show the corresponding results at the paired stations. Some locations, such as in the south of England or the southwest of Australia, exhibit a similar dependence behavior as with the observations, albeit with some bias, while other locations, such as in Italy or the northwest coast of Australia, show opposite results. Nevertheless, we note that at most locations the model can capture the sign of the correlation correctly. Figure S8 indicates that, in general, the selected models tend to correctly identify the locations with the highest co-occurrences but overestimate the number of co-occurring annual maxima. An outlier is the northwest of Australia, where no co-occurring annual maxima are measured based on observations and 10 based on the simulated variables. These discrepancies could be due to the fact that the selected global models fail to capture important small-scale features and processes driving extreme discharge and storm surge at these locations.

[Figure]

**(a) From observation stations**

[Figure]

**(b) From the simulated variables**

Figure S6: Spearman's rank correlation between the storm surge conditional on annual maxima discharge obtained from observation stations (a), and from the simulated variables at these locations (b) .

[Figure]

**(a) From observation stations**

[Figure]

**(b) From the simulated variables**

**Figure S7: Spearman's rank correlation between discharge conditional on annual maxima storm surge obtained from observation stations (a), and from the simulated variables at these locations (b) .**

[Figure]

[Figure]

(b)   From the simulated variables

Figure S8: Number of co-occuring annual maxima of discharge and storm surge from observation stations (a), and from the simulated variables (b) .

**Supplement S2. Co-occurrences of joint annual maxima at selected locations**

[Figure]

**Figure S9: Examples of pseudo-observations from simulated annual maxima of discharge Q and storm surge S at selected locations. Red dots indicate a co-occurrence of Q and S, $(Q^*, S^*)$, within a time lag of 3 days.**

**Supplement S3. Sensitivity of time window on Spearman's $r_s$ correlation coefficient**

| Time window Δ (days) | Storm surge conditional on discharge annual maxima $(Q_n, s_n)$ | | Discharge conditional on storm surge annual maxima $(S_n, q_n)$ | |
|---|---|---|---|---|
| | α = 0.05 | α = 0.10 | α = 0.05 | α = 0.10 |
| 0 | 0.11 | 0.177 | 0.198 | 0.254 |
| 1 | 0.14144 | 0.19 | 0.19 | 0.24 |
| 2 | 0.14 | 0.2020 | 0.19 | 0.254 |
| 3 | 0.14 | 0.2120 | 0.2019 | 0.25 |
| 4 | 0.155 | 0.2120 | 0.198 | 0.24 |
| 5 | 0.155 | 0.2121 | 0.198 | 0.24 |
| 6 | 0.15 | 0.21 | 0.199 | 0.24 |
| 7 | 0.166 | 0.2222 | 0.18 | 0.24 |

**Table S1:**  **Total percentage (%)** of paired locations **along the global coastline** with a positive and statistically significant Spearman's rank correlation coefficient both for $(Q_n, s_n)$ and $(S_n, q_n)$ pairs and significance levels of α = 0.05 and α = 0.10.

[Figure]

**Figure S6S10**: Spearman's $r_s$ correlation coefficient between storm surge conditional on discharge annual maxima ($Q_n$, $s_n$) for a time window of $\Delta = 0$ days (top) and $\Delta = 7$ days (bottom). Black dots denote locations with no significant dependence ($\alpha = 0.05$).

[Figure]

Figure S7S11: Spearman's $r_s$ correlation coefficient between storm surge conditional on discharge annual maxima $(Q_n, s_n)$ for a time window of $\Delta = 0$ days (top) and $\Delta = 7$ days (bottom). Black dots denote locations with no significant dependence ($\alpha = 0.10$).

[Figure]

**Figure S8S12:** Spearman's $r_s$ correlation coefficient between discharge conditional on storm surge annual maxima $(S_n, q_n)$ for a time window of $\Delta = 0$ days (top) and $\Delta = 7$ days (bottom). Black dots denote locations with no significant dependence ($\alpha = 0.05$).

[Figure]

Figure S913: Spearman's $r_s$ correlation coefficient between discharge conditional on storm surge annual maxima ($S_n$, $q_n$) for a time window of $\Delta = 0$ days (top) and $\Delta = 7$ days (bottom). Black dots denote locations with no significant dependence ($\alpha = 0.10$).

**Supplement S4. Influence of the time window on the number of co-occurring annual maxima**

[Figure]

Figure S14: Number of co-occurring yearly maxima of storm surge and discharge obtained between 1980-2014 using a time window of Δ = 0 days (top) and Δ = 7 days (bottom).